# SARS-CoV-2 B.1.617.2 Delta variant replication and immune evasion

Petra Mlcochova[1,2,41], Steven A. Kemp[1,2,3,41], Mahesh Shanker Dhar[4,41], Guido Papa[5], Bo Meng[1,2], Isabella A. T. M. Ferreira[1,2], Rawlings Datir[1,2], Dami A. Collier[2,3], Anna Albecka[5], Sujeet Singh[4], Rajesh Pandey[6], Jonathan Brown[7], Jie Zhou[7], Niluka Goonawardane[7], Swapnil Mishra[8], Charles Whittaker[8], Thomas Mellan[8], Robin Marwal[4], Meena Datta[4], Shantanu Sengupta[6], Kalaiarasan Ponnusamy[4], Venkatraman Srinivasan Radhakrishnan[4], Adam Abdullahi[1,2], Oscar Charles[3], Partha Chattopadhyay[6], Priti Devi[6], Daniela Caputo[9], Tom Peacock[8], Chand Wattal[10], Neeraj Goel[10], Ambrish Satwik[10], Raju Vaishya[11], Meenakshi Agarwal[12], The Indian SARS-CoV-2 Genomics Consortium (INSACOG)*, The Genotype to Phenotype Japan (G2P-Japan) Consortium*, The CITIID-NIHR BioResource COVID-19 Collaboration*, Antranik Mavousian[13], Joo Hyeon Lee[13,14], Jessica Bassi[15], Chiara Silacci-Fegni[15], Christian Saliba[15], Dora Pinto[15], Takashi Irie[16], Isao Yoshida[17], William L. Hamilton[2], Kei Sato[18,19], Samir Bhatt[4,20], Seth Flaxman[21], Leo C. James[5], Davide Corti[15], Luca Piccoli[15], Wendy S. Barclay[8], Partha Rakshit[4,41✉], Anurag Agrawal[6,41✉] & Ravindra K. Gupta[1,2,22,41✉]

The B.1.617.2 (Delta) variant of severe acute respiratory syndrome coronavirus 2 (SARS-CoV-2) was first identified in the state of Maharashtra in late 2020 and spread throughout India, outcompeting pre-existing lineages including B.1.617.1 (Kappa) and B.1.1.7 (Alpha)[1]. In vitro, B.1.617.2 is sixfold less sensitive to serum neutralizing antibodies from recovered individuals, and eightfold less sensitive to vaccine-elicited antibodies, compared with wild-type Wuhan-1 bearing D614G. Serum neutralizing titres against B.1.617.2 were lower in ChAdOx1 vaccinees than in BNT162b2 vaccinees. B.1.617.2 spike pseudotyped viruses exhibited compromised sensitivity to monoclonal antibodies to the receptor-binding domain and the amino-terminal domain. B.1.617.2 demonstrated higher replication efficiency than B.1.1.7 in both airway organoid and human airway epithelial systems, associated with B.1.617.2 spike being in a predominantly cleaved state compared with B.1.1.7 spike. The B.1.617.2 spike protein was able to mediate highly efficient syncytium formation that was less sensitive to inhibition by neutralizing antibody, compared with that of wild-type spike. We also observed that B.1.617.2 had higher replication and spike-mediated entry than B.1.617.1, potentially explaining the B.1.617.2 dominance. In an analysis of more than 130 SARS-CoV-2-infected health care workers across three centres in India during a period of mixed lineage circulation, we observed reduced ChAdOx1 vaccine effectiveness against B.1.617.2 relative to non-B.1.617.2, with the caveat of possible residual confounding. Compromised vaccine efficacy against the highly fit and immune-evasive B.1.617.2 Delta variant warrants continued infection control measures in the post-vaccination era.

India's first wave of SARS-CoV-2 infections in mid-2020 was relatively mild and was controlled by a nationwide lockdown. Following the easing of restrictions, India has seen expansion in cases of coronavirus disease 2019 since March 2021 with widespread fatalities and a death toll of more than 400,000. Cases of the B.1.1.7 Alpha variant, introduced by travel from the UK in late 2020, expanded in the north of India, and it

[1]Cambridge Institute of Therapeutic Immunology & Infectious Disease (CITIID), Cambridge, UK. [2]Department of Medicine, University of Cambridge, Cambridge, UK. [3]University College London, London, UK. [4]National Centre for Disease Control, Delhi, India. [5]MRC – Laboratory of Molecular Biology, Cambridge, UK. [6]CSIR Institute of Genomics and Integrative Biology, Delhi, India. [7]Department of Infectious Diseases, Imperial College London, London, UK. [8]Medical Research Council (MRC) Centre for Global Infectious Disease Analysis, Jameel Institute, School of Public Health, Imperial College London, London, UK. [9]NIHR Bioresource, Cambridge, UK. [10]Sri Ganga Ram Hospital, New Delhi, India. [11]Indraprastha Apollo Hospital, New Delhi, India. [12]Northern Railway Central Hospital, New Delhi, India. [13]Wellcome-MRC Cambridge Stem Cell Institute, Cambridge, UK. [14]Department of Physiology, Development and Neuroscience, University of Cambridge, Cambridge, UK. [15]Humabs Biomed SA, a subsidiary of Vir Biotechnology, Bellinzona, Switzerland. [16]Institute of Biomedical and Health Sciences, Hiroshima University, Hiroshima, Japan. [17]Tokyo Metropolitan Institute of Public Health, Tokyo, Japan. [18]Division of Systems Virology, The Institute of Medical Science, The University of Tokyo, Tokyo, Japan. [19]CREST, Japan Science and Technology Agency, Saitama, Japan. [20]Section of Epidemiology, Department of Public Health, University of Copenhagen, Copenhagen, Denmark. [21]Department of Computer Science, University of Oxford, Oxford, UK. [22]Africa Health Research Institute, Durban, South Africa. [41]These authors contributed equally: Petra Mlcochova, Steven Kemp, Mahesh Shanker Dhar, Partha Rakshit, Anurag Agrawal, Ravindra K. Gupta. *A list of authors and their affiliations appears at the end of the paper. ✉e-mail: partho_rakshit@yahoo.com; a.agrawal@igib.in; rkg20@cam.ac.uk

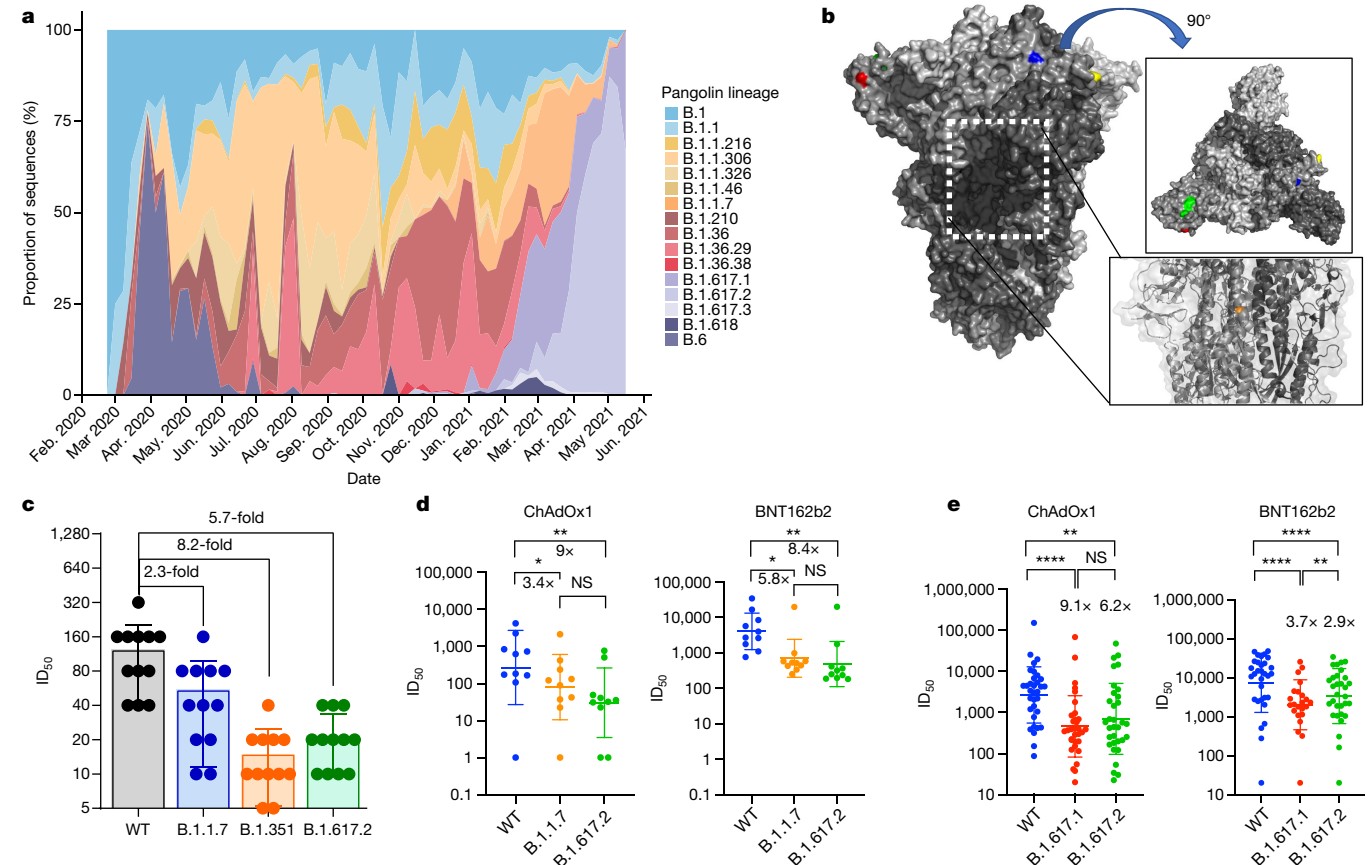

**Fig. 1 | Rapid expansion of Delta variant B.1.617.2 cases in India and reduced sensitivity to neutralizing antibodies from sera derived following infection and vaccination. a**, Proportion of lineages in incident cases of SARS-CoV-2 in India 2020–2021. **b**, Surface representation of the SARS-CoV-2 B.1.671.2 spike trimer (PDB: 6ZGE). Red, L19R; green, del157/158; blue, L452R; yellow, T478K. The white dotted box indicates the location of the D950N substitution (orange). **c**, Neutralization of the Delta variant by convalescent human serum from mid-2020. Fold change in serum neutralization of 100 $TCID_{50}$ of B.1.17 (Alpha), B.1.351 (Beta) and B.1617.2 (Delta) variants relative to WT (IC19); $n = 12$. Shown is the $ID_{50}$, the serum dilution required for 50% virus inhibition, expressed as GMT (from technical replicates) with s.d. **d**, Neutralization of B.1617.2 live virus by sera from vaccinated individuals ($n = 10$ ChAdOx1 or $n = 10$ BNT12b2), compared with B.1.1.7 and Wuhan-1 WT. The graph presents the average of two independent experiments. **e**, Neutralization of B.1.617 spike PV and WT (Wuhan-1 D614G) by vaccine sera ($n = 33$ ChAdOx1 or $n = 32$ BNT162b2). The data are representative of two independent experiments each with two technical replicates. *$P < 0.05$, **$P < 0.01$, ****$P < 0.0001$ (Wilcoxon matched-pairs signed rank test); NS, not significant.

is known to be more transmissible than previous versions of the virus bearing the D614G spike substitution, while maintaining sensitivity to vaccine-elicited neutralizing antibodies[2,3]. The B.1.617 variant was first identified in the state of Maharashtra in late 2020[4], spreading throughout India and to at least 90 countries.

The first sublineage to be detected was B.1.617.1 (ref. [1]), followed by B.1.617.2, both bearing the L452R spike receptor-binding motif (RBM) substitution also observed in B.1.427/B.1.429 (refs. [1,5]). This alteration was previously reported to confer increased infectivity and a modest loss of susceptibility to neutralizing antibodies[6,7]. The B.1.617.2 Delta variant has since dominated over B.1.617.1 (Kappa variant) and other lineages including B.1.1.7, although the reasons remain unclear.

## Delta variant and neutralizing antibodies

We first plotted the relative proportion of variants in new cases of SARS-CoV-2 in India since the start of 2021. Although B.1.617.1 emerged earlier, the Delta variant B.1.617.2 has become more dominant (Fig. 1a). We hypothesized that B.1.617.2 would exhibit immune evasion to antibody responses generated by previous SARS-CoV-2 infection. We used sera from 12 individuals infected during the first UK wave in mid-2020. These sera were tested for their ability to neutralize a B.1.617.2 viral isolate, in comparison with a B.1.1.7 variant isolate and a wild-type (WT) Wuhan-1 virus bearing D614G in spike. The Delta variant contains several spike

alterations that are located at positions within the structure that are predicted to alter its function (Fig. 1b). We found that the B.1.1.7 virus isolate was 2.3-fold less sensitive to the sera than the WT, and that B.1.617.2 was 5.7-fold less sensitive to the sera (Fig. 1c). Importantly, in the same assay, the B.1.351 Beta variant that was first identified in South Africa demonstrated an 8.2-fold loss of neutralization sensitivity relative to the WT.

We used the same B.1.617.2 live virus isolate to test susceptibility to vaccine-elicited serum neutralizing antibodies in individuals following vaccination with two doses of ChAdOx1 or BNT162b2. These experiments showed a loss of sensitivity for B.1.617.2 compared with WT Wuhan-1 bearing D614G of around eightfold for both sets of vaccine sera and reduction against B.1.1.7 that did not reach statistical significance (Fig. 1d). We also used a pseudotyped virus (PV) system to test the neutralization potency of a larger panel of 65 vaccine-elicited sera, this time against B.1.617.1 as well as B.1.617.2 spike compared with Wuhan-1 D614G spike (Fig. 1e). Comparison of demographic data for vaccinees showed similar characteristics (Extended Data Table 1). The mean geometric mean titre (GMT) against Delta variant spike PV was lower for ChAdOx1 than for BNT162b2 (GMT 654 versus 3,372, $P < 0001$, Extended Data Table 1).

We investigated the role of the B.1.617.2 spike as an escape mechanism by testing 33 spike-specific monoclonal antibodies with an in vitro PV neutralization assay using Vero E6 target cells expressing transmembrane protease serine 2 (TMPRSS2) and the Wuhan-1 D614G SARS-CoV-2

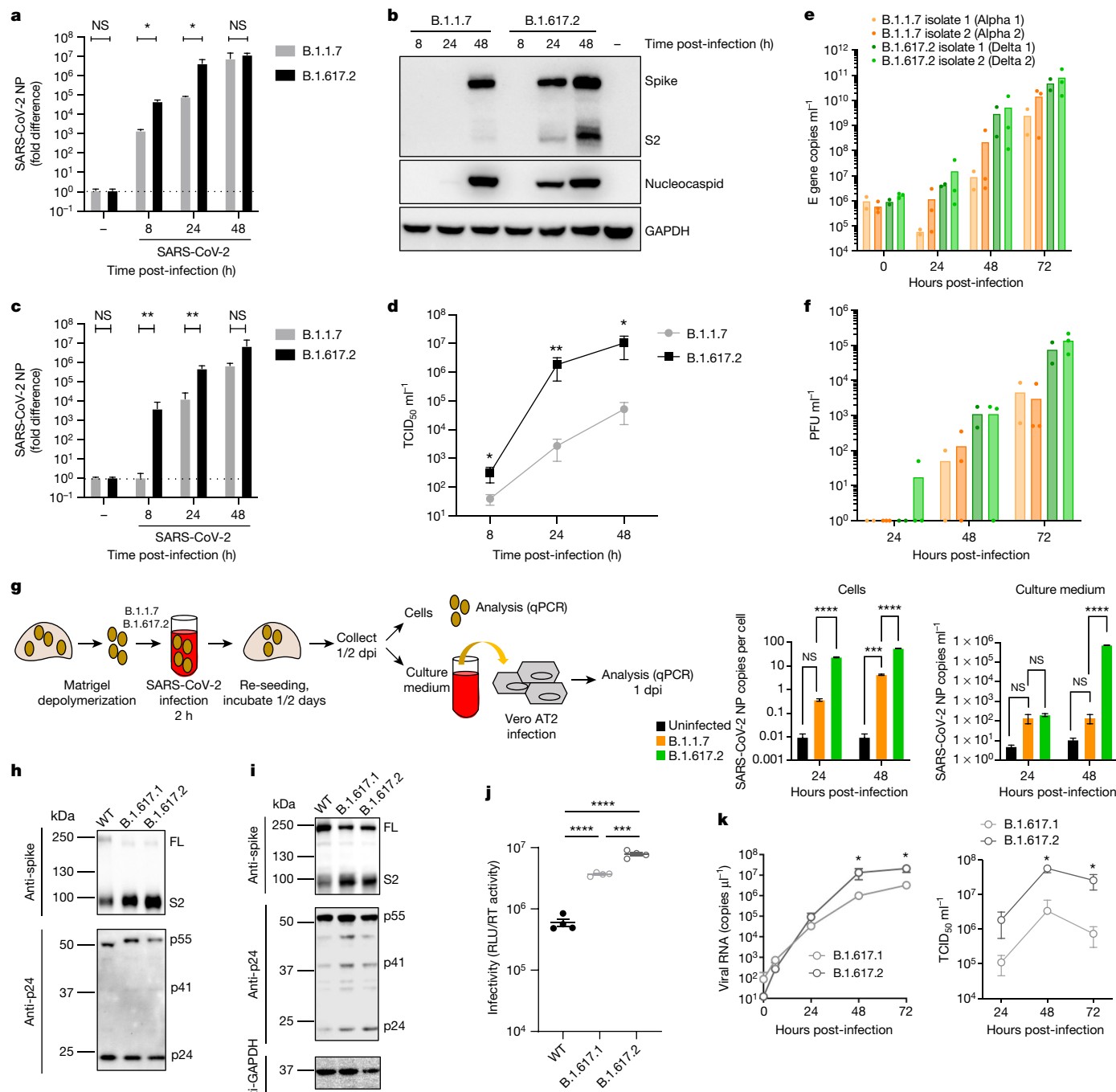

**Fig. 2 | Delta variant live virus replication kinetics and spike-mediated infectivity. a**–**d**, Live virus replication comparing B.1.1.7 with B.1.617.2. Calu-3 cells were infected with variants at an MOI of 0.1. **a**, Viral loads measured by qPCR in cell lysates. **b**, Viral protein levels in cell lysates. **c**, **d**, Live virus produced from infected Calu-3 cell supernatants was collected and used to infect permissive Vero E6 ACE2/TMPRSS2 cells to measure viral loads (**c**) or $TCID_{50}$ $ml^{-1}$ (**d**). **e**, **f**, Virus replication kinetics in the HAE system. **g**, Live virus replication in airway epithelial organoid cultures. Airway epithelial organoids were infected with the SARS-CoV-2 variants B.1.1.7 and B.1.617.2 at an MOI of 1. Cells were lysed and total RNA was isolated. qPCR was used to determine the number of copies of the nucleoprotein gene in cells and the infectivity of cell-free virus measured by infection of Vero E6 ACE2/TMPRSS2 cells. The data are representative of two independent experiments. dpi, days post-infection.

**h**, **i**, Western blots of PV virions (**h**) and cell lysates (**i**) of 293T producer cells following transfection with plasmids expressing lentiviral vectors and SARS-CoV-2 S B.1.617.1 and Delta variant B.1.617.2 versus WT (Wuhan-1 with D614G), probed with antibodies to HIV-1 p24 and SARS-Cov-2 S2. **j**, Calu-3 cell entry by spike B.1.617.2 and B.1.617.1 versus WT D614G parental plasmid PVs. The data are representative of three independent experiments. **k**, Growth kinetics of B.1.617.1 and B.1.617.2 variants. Viral isolates of B.1.617.1 and B.1.617.2 were inoculated into Calu-3 cells, and viral RNA in the culture supernatant was quantified by real-time RT–PCR. The $TCID_{50}$ of released virus in supernatant was measured over time. Assays were performed in quadruplicate. NS, not significant; *$P < 0.05$, **$P < 0.01$, ***$P < 0.001$, ****$P < 0.0001$. The data are representative of two independent experiments. Uninfected cells are represented by a minus symbol. NP, nucleocapsid protein.

spike or the B.1.617.2 spike (Extended Data Fig. 1 and Extended Data Table 2). We found that all three amino-terminal domain monoclonal antibodies (100%) and four out of nine (44%) non-RBM monoclonal

antibodies completely lost neutralizing activity against B.1.617.2. Within the RBM-binding group, 16 out of 26 monoclonal antibodies (61.5%) showed a marked decrease (2- to 35-fold-change reduction) or complete

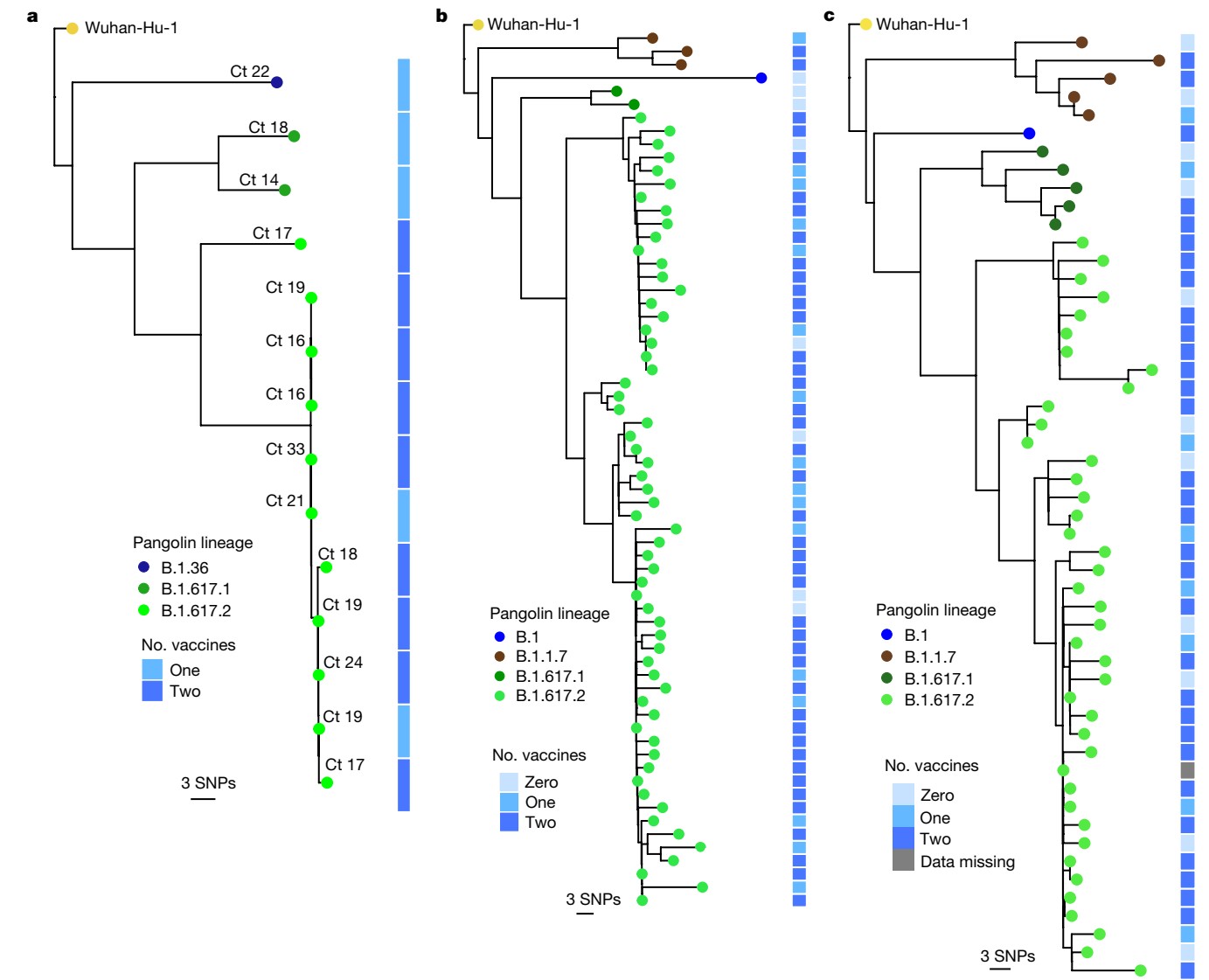

**Fig. 3 | SARS-CoV-2 B.1.617.2 infection in vaccinated HCWs.**
**a**–**c**, Maximum-likelihood phylogenies of vaccine breakthrough SARS-CoV-2
sequences among vaccinated HCWs at three centres. Phylogenies were
inferred with IQTREE2 with 1,000 bootstrap replicates. SNPs, single nucleotide
polymorphisms.

loss (>40-fold-change reduction) of neutralizing activity to B.1.617.2
(Extended Data Fig. 1). Among five clinical-stage RBM monoclonal anti-
bodies tested, bamlanivimab did not neutralize B.1.617.2. Imdevimab,
part of the REGN-COV2 therapeutic dual antibody cocktail[8], displayed
reduced neutralizing activity (Extended Data Fig. 1).

## SARS-CoV-2 Delta variant replication

We first infected a lung epithelial cell line, Calu-3, comparing B.1.1.7 and
B.1.617.2 (Fig. 2a–d). We observed a replication advantage for B.1.617.2
(Fig. 2a, b), as well as an increase in released virions from cells (Fig. 2c,
d). Next we tested B.1.1.7 against two separate isolates of B.1.617.2 in a
human airway epithelial (HAE) model[9]. In this system we again observed
that both B.1.617.2 isolates had a significant replication advantage
over B.1.1.7 (Fig. 2e, f). Finally, we infected primary three-dimensional
airway organoids[10] (Fig. 2g) with B.1.617.2 and B.1.1.7 virus isolates,
noting a significant replication advantage for B.1.617.2 over B.1.1.7.
These data clearly support the higher replication rate and therefore
transmissibility of B.1.617.2 over B.1.1.7.

In the aforementioned experiments, we noted a higher proportion of
intracellular B.1.617.2 spike in the cleaved state, compared with B.1.1.7

(Fig. 2b). We next ran western blots on purified virions probing for
spike S2 and nucleoprotein, revealing B.1.617.2 spike predominantly in
the cleaved form, in contrast to that in B.1 and B.1.1.7 (Extended Data
Fig. 2a, b).

## B.1.617.2 spike–mediated cell fusion

The plasma membrane route of entry, and indeed transmissibility in
animal models, is critically dependent on the polybasic cleavage site
between S1 and S2 (refs. [9,11,12]) and cleavage of spike before virion release
from producer cells. Alterations at P681 in the polybasic cleavage site
have been observed in multiple SARS-CoV-2 lineages, most notably in
the B.1.1.7 Alpha variant[13]. We previously showed that B.1.1.7 spike, bear-
ing P681H, had significantly higher fusogenic potential than a D614G
Wuhan-1 virus[13]. Here we tested B.1.617.1 and B.1.617.2 spike using a split
GFP system to monitor cell–cell fusion (Extended Data Fig. 2c–g). The
B.1.617.1 and B.1.617.2 spike proteins mediated higher fusion activity and
syncytium formation than WT, probably mediated by P681R (Extended
Data Fig. 2f, g). We next titrated sera from ChAdOx1 vaccinees and
showed that indeed the cell–cell fusion could be inhibited in a manner
that mirrored the neutralization activity of the sera against PV infection

of cells (Extended Data Fig. 2h). Hence, B.1.617.2 may induce cell–cell fusion in the respiratory tract and possibly higher pathogenicity even in vaccinated individuals with neutralizing antibodies.

## B.1.617.2 spike–mediated cell entry

We tested single-round viral entry of B.1.617.1 and B.1.617.2 spike (Fig. 2h, i and Extended Data Fig. 3a, b) using the PV system, infecting Calu-3 lung cells expressing endogenous levels of angiotensin-converting enzyme 2 (ACE2) and TMPRSS2 (Fig. 2j), as well as other cells transduced or transiently transfected with ACE2 and TMPRSS2 (Extended Data Fig. 3b). B.1.617 spike proteins were present predominantly in the cleaved form, in contrast to WT (Fig. 2h, i and Extended Data Fig. 3c). We observed 1 log increased entry efficiency for both B.1.617.1 and B.1.617.2 over WT (Extended Data Fig. 3b).

The B.1.617.1 variant was detected before B.1.617.2 in India, and the reasons for B.1.617.2 outcompeting B.1.617.1 are unknown. B.1.617.2 had an entry advantage compared with B.1.617.1 in Calu-3 cells bearing endogenous receptors (Fig. 2j). We confirmed higher infectivity of B.1.617.2 using live virus isolates in Calu-3 cells (Fig. 2k), offering a parsimonious explanation for the epidemiologic growth advantage of B.1.617.2.

## B.1.617.2 vaccine breakthrough infection

We hypothesized that vaccine effectiveness against B.1.617.2 would be compromised relative to that against other circulating variants. Vaccination of health care workers (HCWs) started in early 2021 with the ChAdOx1 vaccine (Covishield). During the wave of infections in March and April, symptomatic SARS-CoV-2 was confirmed in 30 vaccinated staff members among a workforce of 3,800 at a single tertiary centre in Delhi. Genomic data from India and Delhi suggested B.1.1.7 dominance (Fig. 1a and Extended Data Fig. 4a), with growth of B.1.617 during March 2021. Short-read sequencing[14] of symptomatic non-fatal infections in the HCW outbreak revealed that the majority were B.1.617.2 with a range of other B lineage viruses (Fig. 3a). Phylogenetic analysis demonstrated a group of highly related, and in some cases, genetically indistinct sequences that were sampled within 1 or 2 days of each other (Fig. 3a and Extended Data Fig. 4b). We next looked in greater detail at the vaccination history of affected individuals. Nearly all had received two doses at least 21 days previously. We obtained similar data on vaccine breakthrough infections in two other health facilities in Delhi with 1,100 and 4,000 HCW staff members, respectively (Fig. 3b, c and Extended Data Fig. 4c, d). In hospital 2, there were 118 sequences, representing more than 10% of the workforce over a 4-week period. After filtering, we reconstructed phylogenies using 66 with high-quality whole-genome coverage >95%. In hospital 3, there were 70 symptomatic infections, with 52 high-quality genomes used for inferring phylogenies after filtering.

Across the three centres, we noted that the median age and duration of infection of those infected with B.1.617.2 versus non-B.1.617.2 were similar (Extended Data Table 3), with no evidence that B.1.617.2 was associated with higher risk of hospitalization (Extended Data Table 3). Next we evaluated the effect of B.1.617.2 on vaccine effectiveness against symptomatic infection in the HCWs, compared with other lineages. We used multivariable logistic regression to estimate the odds ratio of testing positive with B.1.617.2 versus non-B.1.617.2 in vaccinated relative to unvaccinated individuals[15], adjusting for age, sex and hospital. The adjusted odds ratio for B.1.617.2 relative to non-B.1.617.2 was 5.45 (95% confidence interval 1.39–21.4, $P = 0.018$) for two vaccine doses (Extended Data Table 4).

## Discussion

Here we have combined in vitro experimentation and molecular epidemiology to propose that increased replication fitness and reduced sensitivity of SARS-CoV-2 B.1.617.2 to neutralizing antibodies have contributed to the recent rapid increase of B.1.617.2, compared with B.1.1.7 and other lineages such as B.1.617.1, despite high vaccination rates in adults and/or high prevalence of prior infection[16]. These data are consistent with modelling analyses that support combination of immune evasion and higher transmissibility as likely drivers of the increase in Delta in Delhi[17].

We demonstrate evasion of neutralizing antibodies by a B.1.617.2 live virus with sera from convalescent patients, as well as sera from individuals vaccinated with two different vaccines, one based on an adenovirus vector (ChAdOx1) and the other mRNA based (BNT162b2). The reduced efficacy for imedevimab against B.1.617.2 shown here could translate to compromised clinical efficacy or possible selection of escape variants where there is immune compromise and chronic SARS-CoV-2 infection with B.1.617.2 (ref. [18]).

It is important to consider that increased infectivity at mucosal surfaces and cell–cell fusion and spread[19] may also facilitate 'evasion' from antibodies[20]. Indeed, our work also shows that B.1.617.2 had a fitness advantage compared with B.1.1.7 across physiologically relevant systems including HAE and three-dimensional airway organoids[10] where cell-free and cell–cell infection are likely to be occurring together. These data support the notion of higher infectiousness of B.1.617.2, either due to higher viral burden or higher particle infectivity, resulting in higher probability of person-to-person transmission. We noted that B.1.617.2 live virus particles contained a higher proportion of cleaved spike than B.1.1.7, and postulated that this is involved in the mechanism of increased infectivity. This hypothesis was supported by our observation that PV particles bearing B.1.617.2 spike demonstrated significantly enhanced entry into a range of target cells.

Finally, we report ChAdOx1 vaccine breakthrough infections in HCWs at three Delhi hospitals, demonstrating reduced vaccine effectiveness against B.1.617.2. Therefore, strategies to boost vaccine responses against variants are warranted and attention to infection control procedures is needed in the post-vaccination era.

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

**The Indian SARS-CoV-2 Genomics Consortium (INSACOG)**

Himanshu Chauhan[4], Tanzin Dikid[4], Hema Gogia[4], Hemlata Lall[4], Kalaiarasan Ponnusamy[4], Kaptan Verma[4], Mahesh Shanker Dhar[4], Manoj K. Singh[4], Meena Datta[4], Namita Soni[4], Namonarayan Meena[4], Preeti Madan[4], Priyanka Singh[4], Ramesh Sharma[4], Rajeev Sharma[4], Sandhya Kabra[4], Sattender Kumar[4], Swati Kumari[4], Uma Sharma[4], Urmila Chaudhary[4], Sridhar Sivasubbu[6], Vinod Scaria[6], Chand Wattal[10], J. K. Oberoi[10], Reena Raveendran[10], S. Datta[10], Saumitra Das[23], Arindam Maitra[23], Sreedhar Chinnaswamy[23], Nidhan Kumar Biswas[23], Ajay Parida[24], Sunil K. Raghav[24], Punit Prasad[24], Apurva Sarin[25], Satyajit Mayor[25], Uma Ramakrishnan[25], Dasaradhi Palakodeti[25], Aswin Sai Narain Seshasayee[25], K. Thangaraj[26], Murali Dharan Bashyam[26], Ashwin Dalal[26], Manoj Bhat[27], Yogesh Shouche[27], Ajay Pillai[27], Priya Abraham[28], Varsha Atul Potdar[28], Sarah S. Cherian[28], Anita Sudhir Desai[29], Chitra Pattabiraman[29], M. V. Manjunatha[29], Reeta S. Mani[29], Gautam Arunachal Udupi[29], Vinay Nandicoori[30], Karthik Bharadwaj Tallapaka[30] & Divya Tej Sowpati[30]

[23]National Institute of Biomedical Genomics, Kalyani, India. [24]Institute of Life Sciences (ILS), Bhubaneswar, India. [25]InSTEM/ NCBS, Bangalore, India. [26]Centre for DNA Fingerprinting and Diagnostics, Hyderabad, India. [27]National Centre for Cell Science, Pune, India. [28]National Institute of Virology, Pune, India. [29]National Institute of Mental Health and Neuroscience, Bangalore, India. [30]Centre for Cellular and Molecular Biology, Hyderabad, India.

**The Genotype to Phenotype Japan (G2P-Japan) Consortium**

Ryoko Kawabata[16], Nanami Morizako[16], Kenji Sadamasu[17], Hiroyuki Asakura[17], Mami Nagashima[17], Kazuhisa Yoshimura[17], Jumpei Ito[18], Izumi Kimura[18], Keiya Uriu[18], Yusuke Kosugi[18], Mai Suganami[18], Akiko Oide[18], Miyabishara Yokoyama[18], Mika Chiba[18], Akatsuki Saito[31], Erika P. Butlertanaka[31], Yuri L. Tanaka[31], Terumasa Ikeda[32], Chihiro Motozono[32], Hesham Nasser[32], Ryo Shimizu[32], Yue Yuan[32], Kazuko Kitazato[32], Haruyo Hasebe[32], So Nakagawa[33], Jiaqi Wu[33], Miyoko Takahashi[33], Takasuke Fukuhara[34], Kenta Shimizu[34], Kana Tsushima[34], Haruko Kubo[34], Kotaro Shirakawa[35], Yasuhiro Kazuma[35], Ryosuke Nomura[35], Yoshihito Horisawa[35], Akifumi Takaori-Kondo[35], Kenzo Tokunaga[36] & Seiya Ozono[36]

[31]University of Miyazaki, Miyazaki, Japan. [32]Kumamoto University, Kumamoto, Japan. [33]Tokai University, Tokyo, Japan. [34]Hokkaido University, Sapporo, Japan. [35]Kyoto University, Kyoto, Japan. [36]National Institute of Infectious Diseases, Tokyo, Japan.

**The CITIID-NIHR BioResource COVID-19 Collaboration**

Stephen Baker[1,2], Gordon Dougan[1,2], Christoph Hess[2], Nathalie Kingston[9], Paul J. Lehner[1,2], Paul A. Lyons[1,2], Nicholas J. Matheson[1,2], Willem H. Owehand[22], Caroline Saunders[21], Charlotte Summers[2], James E. D. Thaventhiran[2], Mark Toshner[2], Michael P. Weekes[2], Patrick Maxwell[37], Ashley Shaw[37], Ashlea Bucke[38], Jo Calder[38], Laura Canna[38], Jason Domingo[38], Anne Elmer[38], Stewart Fuller[38], Julie Harris[38], Sarah Hewitt[38], Jane Kennet[38], Sherly Jose[38], Jenny Kourampa[38], Anne Meadows[38], Criona O'Brien[38], Jane Price[38], Cherry Publico[38], Rebecca Rastall[38], Carla Ribeiro[38], Jane Rowlands[38], Valentina Ruffolo[38], Hugo Tordesillas[38], Ben Bullman[1], Benjamin J. Dunmore[2], Stuart Fawke[39], Stefan Gräf[2], Josh Hodgson[4], Christopher Huang[4], Kelvin Hunter[2], Emma Jones[31], Ekaterina Legchenko[2], Cecilia Matara[2], Jennifer Martin[2], Federica Mescia[2], Ciara O'Donnell[2], Linda Pointon[2], Nicole Pond[2], Joy Shih[2], Rachel Sutcliffe[2], Tobias Tilly[2], Carmen Treacy[2], Zhen Tong[2], Jennifer Wood[2], Marta Wylot[2], Laura Bergamaschi[2], Ariana Betancourt[2], Georgie Bower[2], Chiara Cossetti[2], Aloka De Sa[2], Madeline Epping[2], Stuart Fawke[2], Nick Gleadall[2], Richard Grenfell[2], Andrew Hinch[2], Oisin Huhn[39], Sarah Jackson[2], Isobel Jarvis[2], Ben Krishna[2], Daniel Lewis[4], Joe Marsden[4], Francesca Nice[40], Georgina Okecha[3], Ommar Omarjee[2], Marianne Perera[2], Martin Potts[2], Nathan Richoz[2], Veronika Romashova[2], Natalia Savinykh Yarkoni[2], Rahul Sharma[4], Luca Stefanucci[2], Jonathan Stephens[22], Mateusz Strezlecki[2], Lori Turner[2], Eckart M. D. D. De Bie[2], Katherine Bunclark[2], Masa Josipovic[2], Michael Mackay[2], Federica Mescia[2], Sabrina Rossi[37], Mayurun Selvan[4], Sarah Spencer[15], Cissy Yong[37], John Allison[9], Helen Butcher[9,40], Daniela Caputo[9,40], Debbie Clapham-Riley[9,40], Eleanor Dewhurst[9,40], Anita Furlong[9,40], Barbara Graves[9,40], Jennifer Gray[9,40], Tasmin Ivers[9,40], Mary Kasanicki[9,30], Emma Le Gresley[9,40], Rachel Linger[9,40], Sarah Meloy[9,40], Francesca Muldoon[9,40], Nigel Ovington[9], Sofia Papadia[9,40], Isabel Phelan[9,40], Hannah Stark[9,40], Kathleen E. Stirrups[12,22], Paul Townsend[40], Neil Walker[40], Jennifer Webster[9,40], Ingrid Scholtes[40], Sabine Hein[40] & Rebecca King[40]

[37]Cambridge University Hospitals NHS Trust, Cambridge, UK. [38]Cambridge Clinical Research Centre, NIHR Clinical Research Facility, Cambridge University Hospitals NHS Foundation Trust, Addenbrooke's Hospital, Cambridge, UK. [39]Department of Biochemistry, University of Cambridge, Cambridge, UK. [40]University of Cambridge, Cambridge Biomedical Campus, Cambridge, UK.

## Methods

### Serum samples and ethical approval

Ethical approval for the study of vaccine-elicited antibodies in sera from vaccinees was obtained from the East of England – Cambridge Central Research Ethics Committee Cambridge (REC ref. 17/EE/0025). Use of convalescent sera had ethical approval from the South Central - Berkshire B Research Ethics Committee (REC ref. 20/SC/0206; IRAS 283805). Studies involving HCWs (including testing and sequencing of respiratory samples) were reviewed and approved by The Institutional Human Ethics Committees of the National Centre for Disease Control (NCDC) and CSIR-IGIB(NCDC/2020/NERC/14 and CSIR-IGIB/ IHEC/2020-21/01). Participants provided informed consent.

### Sequencing quality control and phylogenetic analysis

Three sets of fasta consensus sequences were obtained from three separate hospitals in Delhi, India. Initially, all sequences were concatenated into a multi-fasta file and then aligned to the reference strain MN908947.3 (Wuhan-Hu-1) with mafft v4.487 (ref. [21]) using the --keeplength and --addfragments options. Following this, all sequences were passed through Nextclade v0.15 (https://clades.nextstrain.org/) to determine the number of gap regions. This was noted and all sequences were assigned a lineage with Pangolin v3.1.5 (ref. [22]) and pangoLEARN (dated 15 June 2021). Sequences that could not be assigned a lineage were discarded. After assigning lineages, all sequences with more than 5% N regions were also excluded.

Phylogenies were inferred using maximum likelihood in IQTREE v2.1.4 (ref. [23]) using a GTR + R6 model with 1,000 rapid bootstraps. The inferred phylogenies were annotated in R v4.1.0 using ggtree v3.0.2 (ref. [24]) and rooted on the SARS-CoV-2 reference sequence (MN908947.3). Nodes were arranged in descending order and lineages were annotated on the phylogeny as coloured tips, alongside a heatmap defining the number of ChAdOx1 vaccine doses received by each patient.

### Structural analyses

The PyMOL Molecular Graphics System v.2.4.0 (https://github.com/ schrodinger/pymol-open-source/releases) was used to map the location of the mutations defining the Delta lineage (B.1.617.2) onto the closed-conformation spike protein (PDB: 6ZGE)[25].

### Statistical analyses

**Vaccine breakthrough infections in HCWs.** Descriptive analyses of demographic and clinical data are presented as median and interquartile range or mean and standard deviation (s.d.) when continuous and as frequency and proportion (%) when categorical. The differences in continuous and categorical data were tested using the Wilcoxon rank sum test or $t$-test and chi-square test, respectively. The association between the Ct value and the SARS-CoV-2 variant was examined using linear regression. Variants as the dependent variable were categorized into two groups: B.1.617.2 variant and non-B.1.617.2 variants. The following covariates were included in the model irrespective of confounding: age, sex, hospital and interval between symptom onset and nasal swab PCR testing.

**Vaccine effectiveness.** To estimate vaccine effectiveness for the B.1.617.2 variant relative to non-B.1.617.2 variants, we adopted a recently described approach[15]. This method is based on the premise that if the vaccine is equally effective against B.1.617.2 and non-B.1.617.2 variants, a similar proportion of cases with each variant would be expected in both vaccinated and unvaccinated cases. This approach overcomes the issue of higher background prevalence of one variant over the other. We determined the proportion of individuals with the B.1.617.2 variant relative to all other circulating variants by vaccination status. We then used logistic regression to estimate the odds ratio of testing positive with B.1.617.2 in vaccinated compared with unvaccinated individuals.

The final regression model was adjusted for age as a continuous variable, and sex and hospital as categorical variables. Model sensitivity and robustness to inclusion of these covariates was tested by an iterative process of sequentially adding the covariates to the model and examining the impact on the odd ratios and confidence intervals until the final model was constructed (Extended Data Table 4). The $R^2$ measure, as proposed by McFadden[26], was used to test the fit of different specifications of the same model regression. This is was performed by sequential addition of the variables adjusted for including age, sex and hospital until the final model was constructed. In addition, the absolute difference in the Bayesian information criterion was estimated. The McFadden $R^2$ measure of final model fitness was 0.11, indicating reasonable model fit. The addition of age, sex and hospital in the final regression model improved the measured fitness. However, the absolute difference in the Bayesian information criterion was 13.34 between the full model and the model excluding the adjusting variable, providing strong support for the parsimonious model. The fully adjusted model was nonetheless used as the final model as the sensitivity analyses (Extended Data Table 4) showed robustness to the addition of the covariates.

**Neutralization titre analyses.** The neutralization by vaccine-elicited antibodies after two doses of the BNT162b2 or ChAdOx1 vaccines was determined by infections in the presence of serial dilutions of sera as described below. The $ID_{50}$ for each group was summarized as a GMT, and statistical comparisons between groups were performed with Mann–Whitney or Wilcoxon ranked sign tests. Statistical analyses were performed using Stata v13 and Prism v9.

### PV experiments

**Cells.** HEK 293T CRL-3216, HeLa-ACE2 (gift from James Voss) and Vero CCL-81 cells were maintained in Dulbecco's modified Eagle medium (DMEM) supplemented with 10% fetal calf serum (FCS), 100 U ml$^{-1}$ penicillin and 100 mg ml$^{-1}$ streptomycin. All cells were regularly tested and found to be mycoplasma free. H1299 cells were a gift from Simon Cook. Calu-3 cells were a gift from Paul Lehner. A549 ACE2/TMPRSS2 (ref. [27]) cells were a gift from Massimo Palmerini. Vero E6 ACE2/TMPRSS2 cells were a gift from Emma Thomson.

**PV preparation for testing against vaccine-elicited antibodies and cell entry.** Plasmids encoding the spike protein of SARS-CoV-2 D614 with a carboxy-terminal 19-amino-acid deletion with D614G were used. Mutations were introduced using the QuikChange Lightning Site-Directed Mutagenesis kit (Agilent) following the manufacturer's instructions. Preparation of the B.1.1.7 S-expressing plasmid was described previously, but in brief, it was generated by stepwise mutagenesis. Viral vectors were prepared by transfection of 293T cells by using Fugene HD transfection reagent (Promega). 293T cells were transfected with a mixture of 11 μl Fugene HD, 1 μg pCDNAΔ19 spike–HA, 1 μg p8.91 human immunodeficiency virus 1 (HIV-1) gag–pol expression vector and 1.5 μg pCSFLW (expressing the firefly luciferase reporter gene with the HIV-1 packaging signal). Viral supernatant was collected at 48 and 72 h after transfection, filtered through a 0.45-μm filter and stored at −80 °C as previously described. Infectivity was measured by luciferase detection in target 293T cells transfected with TMPRSS2 and ACE2, Vero E6 ACE2/TMPRSS2, Calu-3, A549 ACE2/TMPRSS2, H1299 and HeLa-ACE2 cells.

**Standardization of virus input by SYBR Green-based product-enhanced PCR assay.** The reverse transcriptase (RT) activity of virus preparations was determined by quantitative PCR (qPCR) using a SYBR Green-based product-enhanced PCR assay as previously described[28]. In brief, tenfold dilutions of virus supernatant were lysed in a 1:1 ratio in a 2× lysis solution (made up of 40% glycerol (v/v), 0.25% Triton X-100 (v/v), 100 mM KCl, RNase inhibitor 0.8 U ml$^{-1}$, Tris HCl 100 mM, buffered to pH 7.4) for 10 min at room temperature.

A 12-µl volume of each sample lysate was added to 13 µl of a SYBR Green master mix (containing 0.5 µM MS2-RNA forward and reverse primers, 3.5 pmol ml$^{-1}$ MS2-RNA and 0.125 U µl$^{-1}$ Ribolock RNAse inhibitor) and cycled in a QuantStudio. Relative amounts of RT activity were determined as the rate of transcription of bacteriophage MS2 RNA, with absolute RT activity calculated by comparing the relative amounts of RT activity with an RT standard of known activity.

### Viral isolate comparison between B.1.617.1 and B.1.617.2

**Cell culture.** Vero E6 TMPRSS2 cells (an African green monkey (*Chlorocebus sabaeus*) kidney cell line; JCRB1819)[29] were maintained in DMEM (low glucose) (Wako, catalogue no. 041-29775) containing 10% FCS, G418 (1 mg ml$^{-1}$; Nacalai Tesque, catalogue no. G8168-10ML) and 1% antibiotics (penicillin and streptomycin (P/S)).

Calu-3 cells (a human lung epithelial cell line; ATCC HTB-55) were maintained in minimum essential medium Eagle (Sigma-Aldrich, catalogue no. M4655-500ML) containing 10% FCS and 1% PS.

**SARS-CoV-2 B.1.617.1 versus B.1.617.2 experiment.** Two viral isolates belonging to the B.1.617 lineage, B.1.617.1 (GISAID ID: EPI_ISL_2378733) and B.1.617.2 (GISAID ID: EPI_ISL_2378732), were isolated from SARS-CoV-2-positive individuals in Japan. Briefly, 100 µl of the naso-pharyngeal swab obtained from SARS-CoV-2-positive individuals was inoculated into Vero E6 TMPRSS2 cells in a biosafety level 3 laboratory. After incubation at 37 °C for 15 min, a maintenance medium (Eagle's minimum essential medium (FUJIFILM Wako Pure Chemical Corporation, catalogue no. 056-08385) including 2% FCS and 1% PS) was added, and the cells were cultured at 37 °C under 5% CO$_2$. The cytopathic effect (CPE) was confirmed under an inverted microscope (Nikon), and the viral load of the culture supernatant in which CPE was observed was confirmed by real-time PCR with reverse transcription (RT–PCR). To determine viral genome sequences, RNA was extracted from the culture supernatant using the QIAamp viral RNA mini kit (Qiagen, catalogue no. 52906). A cDNA library was prepared using NEB Next Ultra RNA Library Prep Kit for Illumina (New England Biolab, catalogue no. E7530), and whole-genome sequencing was performed using a Miseq instrument (Illumina).

To prepare the working virus, 100 µl of the seed virus was inoculated into Vero E6 TMPRSS2 cells (5,000,000 cells in a T-75 flask). At 1 h after infection, the culture medium was replaced with DMEM (low glucose) (Wako, catalogue no. 041-29775) containing 2% FBS and 1% PS; at 2–3 days post-infection, the culture medium was collected and centrifuged, and the supernatants were collected as the working virus.

The titre of the prepared working virus was measured as 50% tissue culture infectious dose (TCID$_{50}$). Briefly, 1 day before infection, Vero E6 TMPRSS2 cells (10,000 cells per well) were seeded into a 96-well plate. Serially diluted virus stocks were inoculated onto the cells and incubated at 37 °C for 3 days. The cells were observed by microscopy to judge the CPE appearance. The TCID$_{50}$ ml$^{-1}$ value was calculated with the Reed–Muench method[30].

One day before infection, 20,000 Calu-3 cells were seeded into a 96-well plate. SARS-CoV-2 (200 TCID$_{50}$) was inoculated and incubated at 37 °C for 1 h. The infected cells were washed, and 180 µl of culture medium was added. The culture supernatant (10 µl) was collected at indicated time points and used for real-time RT–PCR to quantify the viral RNA copy number.

**Real-time RT–PCR.** Real-time RT–PCR was performed as previously described[31,32]. In brief, 5 µl of culture supernatant was mixed with 5 µl of 2× RNA lysis buffer (2% Triton X-100, 50 mM KCl, 100 mM Tris HCl (pH 7.4), 40% glycerol, 0.8 U µl$^{-1}$ recombinant RNase inhibitor (Takara, catalogue no. 2313B)) and incubated at room temperature for 10 min. RNase-free water (90 µl) was added, and the diluted sample (2.5 µl) was used as the template for real-time RT–PCR performed according to the manufacturer's protocol using the One Step TB Green PrimeScript

PLUS RT–PCR kit (Takara, catalogue no. RR096A) and the following primers: forward *N*, 5′-AGCCTCTTCTCGTTCCTCATCAC-3′; and reverse *N*, 5′-CCGCCATTGCCAGCCATTC-3′. The copy number of viral RNA was standardized with a SARS-CoV-2 direct detection RT–qPCR kit (Takara, catalogue no. RC300A). The fluorescent signal was acquired using a QuantStudio 3 Real-Time PCR system (Thermo Fisher Scientific), a CFX Connect Real-Time PCR Detection System (Bio-Rad) or a 7500 Real Time PCR System (Applied Biosystems).

**Virus growth kinetics in HAE cells.** Primary nasal HAE cells at the air–liquid interface were purchased from Epithelix and the basal MucilAir medium (Epithelix) was changed every 2–3 days for maintenance of HAE cells. All dilution of viruses, wash steps and collection steps were carried out with OptiPRO serum-free medium (SFM; Life Technologies) containing 2× GlutaMAX (Gibco). All wash and collection steps were performed by addition of 200 µl SFM to the apical surface and incubation for 10 min at 37 °C before removing SFM. To infect cells, the basal medium was replaced, and the apical surface of the HAE cells was washed once with SFM to remove mucus before addition of virus to triplicate wells. Cells were infected at a multiplicity of infection (MOI) of $1 \times 10^4$ genome copies of virus per cell based on E gene qRT–PCR. The cells were incubated with the inoculum for 1 h at 37 °C before washing their apical surface twice and retaining the second wash as the sample for 0 hpi. A single apical wash was performed to collect virus at 24, 48 and 71 h time points. Isolates used were B.1.617.2 isolate no. 60 hCoV-19/England/SHEF-10E8F3B/2021 (EPI_ISL_1731019), B.1.617.2 isolate no. 285 hCoV-19/England/PHEC-3098A2/2021 (EPI_ISL_2741645) and B.1.1.7 isolate no. 7540 SMH2008017540 (confirmed B.1.1.7 in-house but not yet available on GISAID).

**Titration of outputs from HAE infections.** For quantifying genome copies in the virus inputs and in the supernatant collected from HAE cells, RNA was extracted using the QIAsymphony DSP Virus/Pathogen Mini Kit on the QIAsymphony instrument (Qiagen). qRT–PCR was then performed using the AgPath RT–PCR (Life Technologies) kit on a QuantStudio(TM) 7 Flex System with the primers for SARS-CoV-2 E gene used previously[33]. A standard curve was also generated using dilutions of viral RNA of known copy number to allow quantification of E gene copies in the samples from $Ct$ values. E gene copies per millilitre of original virus supernatant were then calculated.

For measuring infectious virus in samples collected from HAE cells, plaque assays were carried out by performing serial log dilutions of supernatant in DMEM, 1% NEAA and 1% P/S and inoculating onto PBS-washed Vero cells, incubating for 1 h at 37 °C, removing inoculum and overlaying with 1× MEM, 0.2% (w/v) BSA, 0.16% (w/v) NaHCO$_3$, 10 mM HEPES, 2 mM L-glutamine, 1× P/S, 0.6% (w/v) agarose. Plates were incubated for 3 days at 37 °C before the overlay was removed and cells were stained for 1 h at room temperature in crystal violet solution.

**Lung organoid infection by replication-competent SARS-CoV-2 isolates.** Airway epithelial organoids were prepared as previously reported[10]. For viral infection, primary organoids were passaged and incubated with SARS-CoV-2 in suspension at an MOI of 1 for 2 h. Subsequently, the infected organoids were washed twice with PBS to remove the viral particles. Washed organoids were plated in 20-µl Matrigel domes, cultured in organoid medium and collected at different time points.

Cells were lysed 24 and 48 h post-infection and total RNA was isolated. cDNA was synthesized and qPCR was used to quantify copies of the nucleoprotein gene in samples. A standard curve was prepared using SARS-CoV-2 Positive Control plasmid encoding full nucleocapsid protein (N gene; NEB) and used to quantify copies of the N gene in organoid samples. 18S ribosomal RNA was used as a housekeeping gene to normalize sample-to-sample variation.

**Western blotting.** Cells were lysed and supernatants were collected 48 h post transfection. Purified virions were prepared by collecting supernatants and passing them through a 0.45-μm filter. Clarified supernatants were then loaded onto a thin layer of 8.4% OptiPrep density gradient medium (Sigma-Aldrich) and placed in a TLA55 rotor (Beckman Coulter) for ultracentrifugation for 2 h at 20,000 r.p.m. The pellet was then resuspended for western blotting. Cells were lysed with cell lysis buffer (Cell Signaling), treated with Benzonase nuclease (70664 Millipore) and boiled for 5 min. Samples were then run on 4–12% Bis Tris gels and transferred onto nitrocellulose or polyvinylidene fluoride membranes using an iBlot or semidry system (Life Technologies and Bio-Rad, respectively).

Membranes were blocked for 1 h in 5% non-fat milk in PBS + 0.1% Tween-20 (PBST) at room temperature with agitation, incubated in primary antibody (anti-SARS-CoV-2 spike, which detects the S2 subunit of SARS-CoV-2 S (Invitrogen, PA1-41165), anti-GAPDH (Proteintech) or anti-p24 (NIBSC)) diluted in 5% non-fat milk in PBST for 2 h at 4 °C with agitation, washed four times in PBST for 5 min at room temperature with agitation and incubated in secondary antibodies anti-rabbit HRP (1:10,000, Invitrogen 31462) and anti-β-actin HRP (1:5,000; sc-47778) diluted in 5% non-fat milk in PBST for 1 h with agitation at room temperature. Membranes were washed four times in PBST for 5 min at room temperature and imaged directly using a ChemiDoc MP imaging system (Bio-Rad).

**Virus infection for virion western blotting.** Vero E6 ACE2/TMPRSS2 cells were infected with an MOI of 1 and incubated for 48 h. Supernatant was cleared by a 5-min spin at $300g$ and then precipitated with 10% PEG6000 (4 h at room temperature). Pellets were resuspended directly in Laemmli buffer with 1 mM dithiothreitol, treated with Benzonase nuclease (70664 Millipore) and sonicated before loading for gel electrophoresis

**Serum pseudotype neutralization assay**
Spike pseudotype assays have been shown to have similar characteristics to neutralization testing using fully infectious WT SARS-CoV-2 ([34]). Virus neutralization assays were performed on 293T cells transiently transfected with ACE2 and TMPRSS2 using SARS-CoV-2 spike PV expressing luciferase[35]. PV was incubated with serial dilutions of heat-inactivated human serum samples or convalescent plasma in duplicate for 1 h at 37 °C. Virus- and cell-only controls were also included. Then, freshly trypsinized 293T ACE2/TMPRSS2-expressing cells were added to each well. Following a 48-h incubation in a 5% $CO_2$ environment at 37 °C, the luminescence was measured using the Steady-Glo Luciferase assay system (Promega).

**Neutralization assays for convalescent plasma.** Convalescent serum samples from HCWs at St Mary's Hospital at least 21 days since PCR-confirmed SARS-CoV-2 infection were collected in May 2020 as part of the REACT2 study.

Convalescent human serum samples were inactivated at 56 °C for 30 min, and replicate serial twofold dilutions ($n = 12$) were mixed with an equal volume of SARS-CoV-2 (100 $TCID_{50}$; total volume 100 μl) at 37 °C for 1 h. Vero E6 ACE2/TMPRSS2 cells were subsequently infected with serial fold dilutions of each sample for 3 days at 37 °C. Virus neutralization was quantified via crystal violet staining and scoring for CPE. Each run included 1:5 dilutions of each test sample in the absence of virus to ensure virus-induced CPE in each titration. Back titrations of SARS-CoV-2 infectivity were performed to demonstrate infection with ~100 $TCID_{50}$ in each well.

**Vaccinee serum neutralization, live virus assays**
Vero E6 ACE2/TMPRSS2 cells were seeded at a cell density of $2 \times 10^4$ per well in a 96-well plate 24 h before infection. Serum was titrated starting at a final 1:10 dilution, with WT (SARS-CoV-2/human/Liverpool/REMRQ0001/2020), B.1.1.7 or B.1.617.2 virus isolates being added at an MOI of 0.01. The mixture was incubated for 1 h before adding to cells. The plates were fixed with 8% PFA 72 h post-infection and stained with Coomassie blue for 20 min. The plates were washed in water and dried for 2 h. 1% SDS solution was then added to wells and the staining intensity was measured using FLUOstar Omega (BMG Labtech). The percentage of cell survival was determined by comparing the intensity of staining with that in an uninfected well. A nonlinear sigmoidal 4PL model (Graphpad Prism 9.1.2) was used to determine the $ID_{50}$ for each serum.

**Vesicular stomatitis virus pseudovirus generation for monoclonal antibody assays**
Replication-defective vesicular stomatitis virus (VSV) pseudovirus expressing SARS-CoV-2 spike proteins corresponding to the different variants of concern were generated as previously described with some modifications[36]. Lenti-X 293T cells (Takara, 632180) were seeded in 10-cm² dishes at a density of $5 \times 10^6$ cells per dish and the following day transfected with 10 μg of WT or B.1.617.2 spike expression plasmid with TransIT-Lenti (Mirus, 6600) according to the manufacturer's instructions. One day post-transfection, cells were infected with VSV–luc (VSV G) with an MOI of 3 for 1 h, rinsed three times with PBS containing $Ca^{2+}$/ $Mg^{2+}$, and then incubated for an additional 24 h in complete medium at 37 °C. The cell supernatant was clarified by centrifugation, filtered (0.45 μm), aliquoted and frozen at −80 °C.

**PV neutralization assay for monoclonal antibody.** Vero E6 cells expressing TMPRSS2 or not were grown in DMEM supplemented with 10% FBS and seeded into white 96-well plates (PerkinElmer, 6005688) at a density of 20 thousand cells per well. The next day, monoclonal antibodies were serially diluted in pre-warmed complete medium, mixed with WT or B.1.617.2 pseudoviruses and incubated for 1 h at 37 °C in round-bottom polypropylene plates. Medium from cells was aspirated and 50 μl of virus–monoclonal antibody complexes was added to cells and then incubated for 1 h at 37 °C. An additional 100 μl of pre-warmed complete medium was then added on top of complexes, and cells were incubated for an additional 16–24 h. Conditions were tested in duplicate wells on each plate and at least six wells per plate contained untreated infected cells (defining the 0% of neutralization, MAX relative light unit (RLU) value) and infected cells in the presence of S2E12 and S2X259 at 25 μg ml⁻¹ each (defining the 100% of neutralization, MIN RLU value). Medium containing virus–monoclonal antibody complexes was then aspirated from cells and 50 μl of a 1:2 dilution of SteadyLite Plus (Perkin Elmer, 6066759) in PBS with $Ca^{2+}$ and $Mg^{2+}$ was added to cells. Plates were incubated for 15 min at room temperature and then analysed on the Synergy-H1 (Biotek). The average RLU value for untreated infected wells ($MAX RLU_{ave}$) was subtracted by the average MIN RLU ($MIN RLU_{ave}$) value and used to normalize the percentage of neutralization of individual RLU values of experimental data according to the following formula: $(1 - (RLU_x - MIN RLU_{ave})/(MAX RLU_{ave} - MIN RLU_{ave})) \times 100$. Data were analysed and visualized with Prism (Version 9.1.0). $IC_{50}$ values were calculated from the interpolated value from the log[inhibitor] versus response, using variable slope (four parameters) nonlinear regression with an upper constraint of ≤100, and a lower constraint equal to 0. Each neutralization assay was conducted on two independent experiments (that is, biological replicates), with each biological replicate containing a technical duplicate. $IC_{50}$ values across biological replicates are presented as arithmetic mean ± s.d. The loss or gain of neutralization potency across spike variants was calculated by dividing the variant $IC_{50}$ by the WT $IC_{50}$ within each biological replicate, and then visualized as arithmetic mean ± s.d.

**Plasmids for split GFP system to measure cell–cell fusion.** pQCXIP-BSR-GFP11 and pQCXIP-GFP1–10 were from Yutaka Hata[37] (Addgene plasmid no. 68716; http://n2t.net/addgene:68716; RRID:Addgene_68716 and Addgene plasmid no. 68715; http://n2t.net/addgene:68715; RRID:Addgene_68715).

**Generation of GFP1–10 or GFP11 lentiviral particles.** Lentiviral particles were generated by co-transfection of Vero cells with pQCXIP-BSR-GFP11 or pQCXIP-GFP1–10 as previously described[38]. Supernatant containing virus particles was collected after 48 and 72 h, 0.45-μm filtered, and used to infect 293T or Vero cells to generate stable cell lines. 293T and Vero cells were transduced to stably express GFP1–10 or GFP11, respectively, and were selected with 2 μg ml$^{-1}$ puromycin.

**Cell–cell fusion assay.** The cell–cell fusion assay was carried out as previously described[38,39] but using a split GFP system. Briefly, Vero GFP1–10 and Vero-GFP11 cells were seeded at 80% confluence in a 1:1 ratio in a 24-well plate the day before. Cells were co-transfected with 0.5 μg of spike expression plasmids in pCDNA3 using Fugene 6 following the manufacturer's instructions (Promega). Cell–cell fusion was measured using an Incucyte and determined as the proportion of green area to total phase area. Data were then analysed using Incucyte software. Graphs were generated using Prism 8 software.

## Reporting summary

Further information on research design is available in the Nature Research Reporting Summary linked to this paper.

## Data availability

All SARS-CoV-2 fasta consensus sequence files used in this analysis are available from https://gisaid.org (with accession numbers: hospital 1, EPI_ISIL_1970102–EPI_ISIL_17010116; hospital 2, EPI_ISIL_2461070–EPI_ISIL_2955768; hospital 3, EPI_ISL_2955782–EPI_ISL_3066853) or https://github.com/Steven-Kemp/hospital_india/tree/main/consensus_fasta. All consensus sequence data have also been submitted to NCBI GenBank and can be found with the accession numbers MZ724413–MZ724540.

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

**Acknowledgements** We thank the Department of Biotechnology, NCDC, for coordination of the HCW aspect of the study. R.K.G. is supported by a Wellcome Trust Senior Fellowship in Clinical Science (WT108082AIA). This study was supported by the Cambridge NIHRB Biomedical Research Centre. We also thank A. Mutreja for discussions. We thank T. de Silva for the Delta isolate and K. Kimelian for discussions. S.A.K. is supported by the Bill and Melinda Gates Foundation via PANGEA grant OPP1175094. I.A.T.M.F. is funded by a SANTHE award (DEL-15-006). S. Flaxman acknowledges the EPSRC (EP/V002910/1). We thank P. Lehner, J. Voss, S. Cook, M. Palmerini and E. Thomson for Calu-3 cells, HeLa ACE2, H1299, A549 ACE2/TMPRSS2 and Vero E6 ACE2/TMPRSS2 cells, respectively. We thank C. Lloyd and S. Saglani for providing the primary airway epithelial cultures . We thank the Geno2pheno UK consortium. We acknowledge support from the G2P-UK National Virology consortium funded by MRC/UKRI (grant ref. MR/W005611/1). This study was also supported by The Rosetrees Trust and the Geno2pheno UK consortium. K. Sato is supported by the AMED Research Program on Emerging and Re-emerging Infectious Diseases (20fk0108270 and 20fk0108413), JST SICORP (JPMJSC20U1 and JPMJSC21U5) and JST CREST (JPMJCR20H4).

**Author contributions** Conceived study: A.A., P.R., S.A.K., D.C., S. Bhatt, S. Flaxman, S. Mishra, R.K.G., K. Sato, D.A.C., L. Piccoli, J.B., D. Pinto, D.C. Designed study and experiments: B.M., P. Mlcochova, R.K.G., J.B., N.G., L.C.J., G.P., K. Sato, I.A.T.M. Performed experiments: P. Mlcochova, B.M., D.A.C., A. Abdullahi, R.D., I.A.T.M.F., G.P., C.S.-F., C. Saliba, R.M., M.D., D. Pinto, T. Irie, I.Y., L.C.G., J.B., J.Z., N.G., G.B.M. Patient data collection and analysis: M.S.D., S.S., R.P., N. Goel, A. Satwik, R.V., M.A., A. Mavousian, J.H.L., P.D., P.C., D.C., S. Sengupta, K.P., V.S.R. Performed bioinformatic analyses: W.L.H., M.S.D., S.A.K., O.C. Performed statistical analyses: D.A.C., S. Flaxman, S. Bhatt, C.W., T.M., S.P. Interpreted data: R.K.G., S.A.K., A.A., S.S., J.B., N.G., R.P., P.C., P.D., K.P., V.S.R., S.S., D.C., T.P., O.C., K. Sato, G.P., T. Irie, I.Y., L.C.J., W.S.B., G.P., S. Flaxman, S. Bhatt, D.A.C., B.M., R.D., I.A.T.M.F., P.R., J.B., K.G.C.S., S. Mishra, C. Wattal, M.S.D., T.M., S. Bhatt, L. Piccoli, D.C., C. Saliba, W.L.H., C.S.-F. and S. Flaxman.

**Competing interests** J.B., C.S.-F., C. Saliba, D. Pinto, D.C. and L. Piccoli are employees of Vir Biotechnology and may hold shares in Vir Biotechnology. R.K.G. has received consulting fees from Johnson and Johnson and GSK. The remaining authors declare no competing interests.

**Additional information**
**Correspondence and requests for materials** should be addressed to Partha Rakshit, Anurag Agrawal or Ravindra K. Gupta.

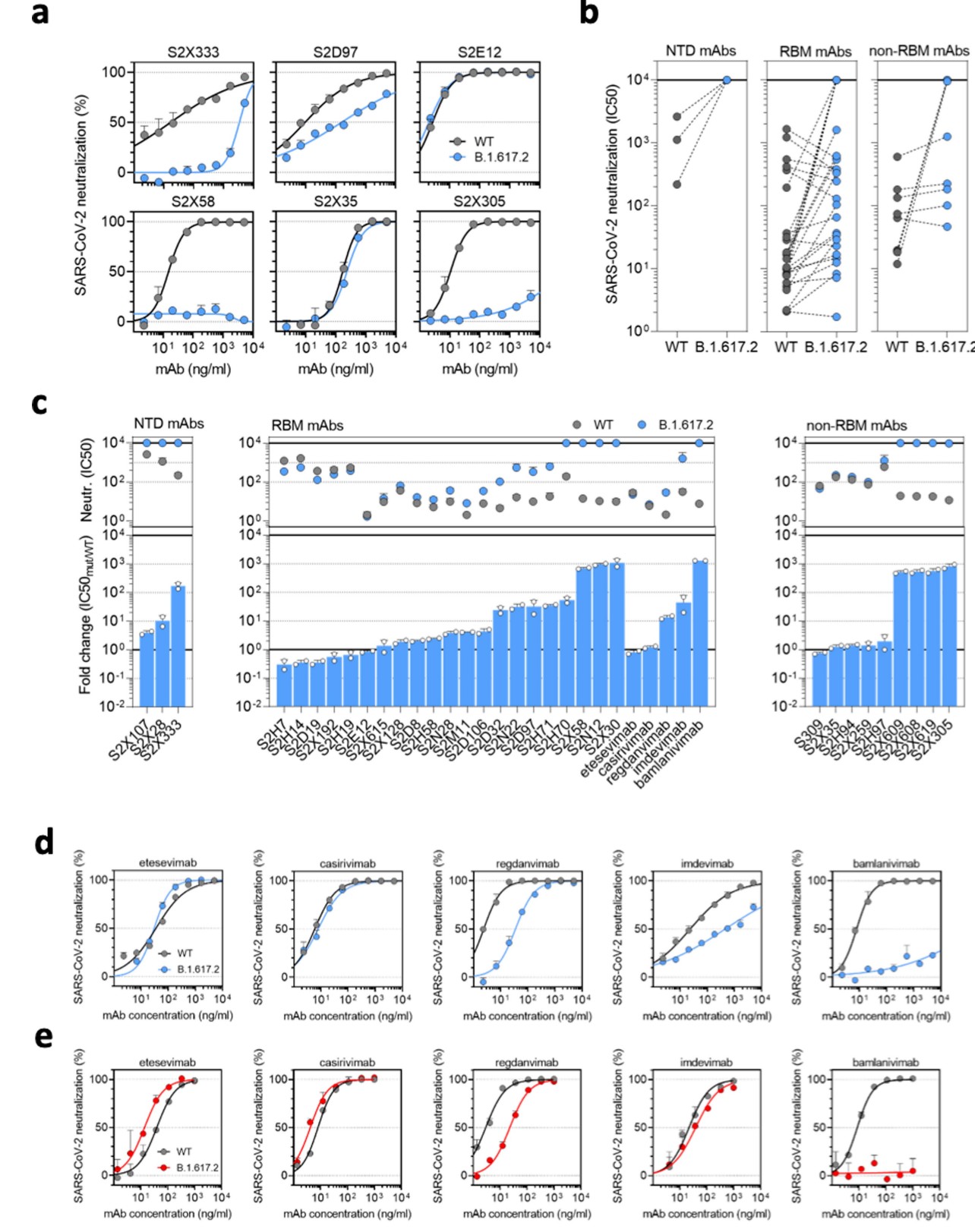

**Extended Data Fig. 1 | See next page for caption.**

**Extended Data Fig. 1 | Delta variant B.1.617.2 shows reduced sensitivity to monoclonal antibodies.** Neutralisation by a panel of NTD- and RBD-specific mAbs against WT and B.1.617.2 mutant SARS-CoV-2 pseudotyped viruses. **a**. Neutralisation of WT D614 (black) and B.1.617.2 mutant (blue) pseudotyped SARS-CoV-2-VSV by 6 selected mAbs from one representative experiment out of 2 independent experiments. S2X333 is an NTD-specific mAb, S2D97, S2E12 and S2X58 are RBM-specific mAbs, while S2X35 and S2X305 are non-RBM mAbs. **b**. Neutralisation of WT and B.1.617.2 VSV by 38 mAbs targeting NTD ($n = 3$), RBM ($n = 26$, including 5 clinical stage mAb) and non-RBM ($n = 9$). Shown are the mean IC50 values (ng/ml) from 2 independent experiments. Non-neutralising IC50 titers were set at $10^4$ ng/ml. **c**. Neutralisation shown as mean IC50 values (upper panel) and average fold change of B.1.617.2 relative to WT (lower panel) of 38 mAbs tested in 2 independent experiments (including 5 clinical-stage mAbs), tested using Vero E6 cells expressing TMPRSS2. **d**–**e**, Neutralisation of WT D614 (black) and B.1.617.2 mutant (blue/red) pseudotyped SARS-CoV-2-VSV by 5 clinical-stage mAbs using Vero E6 cells expressing TMPRSS2 (d) or not (e). Shown is one representative experiment out of 2 independent experiments.

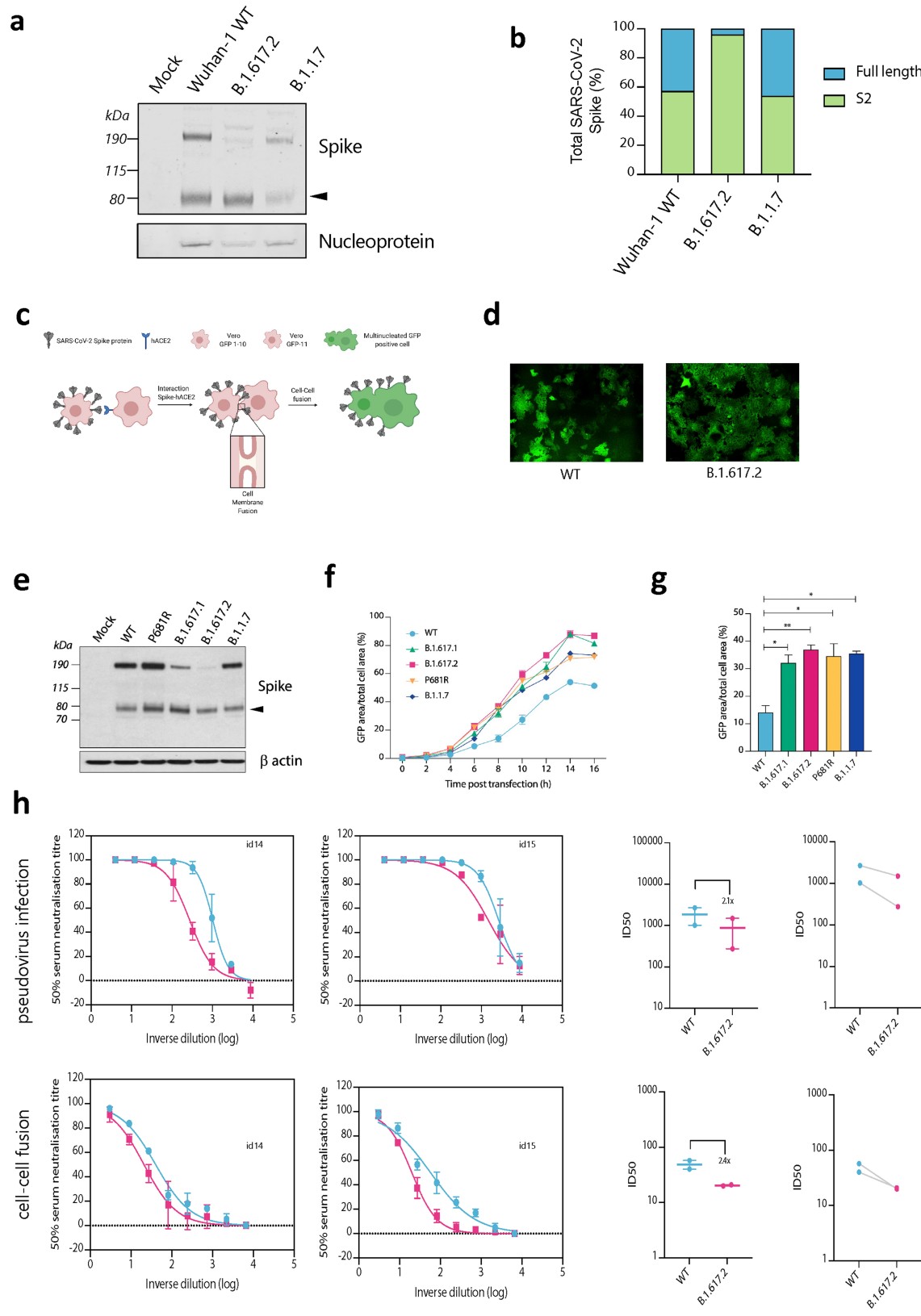

**Extended Data Fig. 2 | Spike cleavage in B.1.617.2 virions compared to B.1.1.7. and spike mediated cell-cell fusion. a**. Representative western blot analysis of spike and nucleoprotein present in SARS-CoV-2 particles from the indicated viruses produced in Vero E6 ACE2/TMPRSS2 cells 48 h post infection. The arrowhead identifies the S2 subunit. **b**. Quantification of cleaved and full-length spike of the indicated viruses. **c**. Schematic of cell-cell fusion assay. **d**. Reconstructed images at 10 h of GFP positive syncytia

formation. Scale bars represent 400 mm. **e**. western blot of cell lysates 48 h after transfection of spike plasmids. **f**,**g**. Quantification of cell-cell fusion kinetics showing percentage of green area to total cell area over time. Mean is plotted with error bars representing SEM. **h**. Comparison of impact of post vaccine sera ($n = 2$) on PV neutralisation (top) and cell-cell fusion (bottom), comparing WT and Delta variant B.1.671.2. Data are representative of at least two independent experiments.

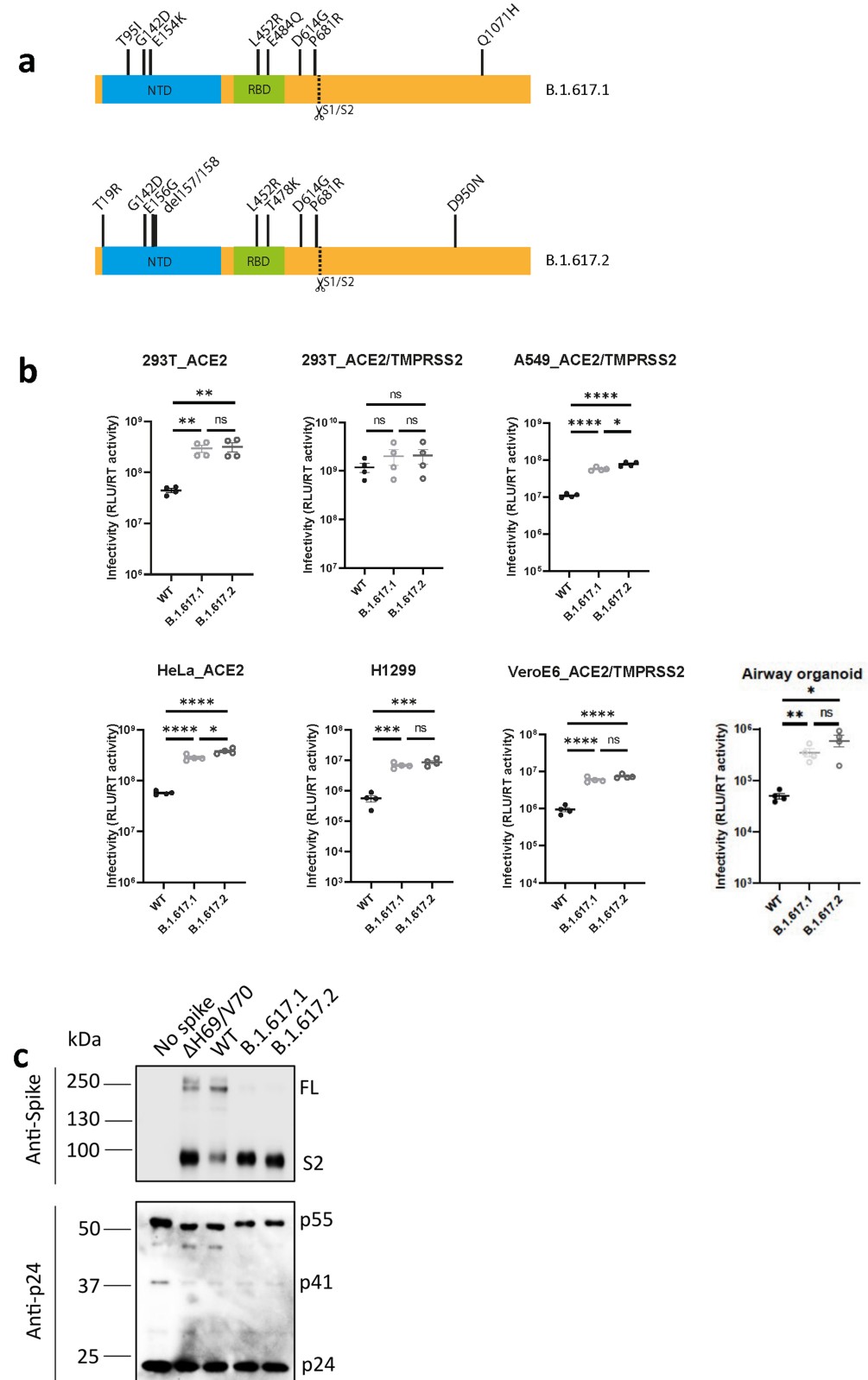

**Extended Data Fig. 3 | B.1.617.2 spike confers increased cell entry.**
**a**. diagram showing mutations present in spike plasmids used for cell entry PV experiments **b**. Single round infectivity on different cell targets by spike B.1.617.1 and B.1.617.1 versus WT (Wuhan-1 D614G) PV produced in 293T cells.

Data are representative of three independent experiments. Statistics were performed using unpaired Student t test. **c**. Western blotting of supernatants from transfected 293T probing for S2 and p24 in PV and showing no spike control.

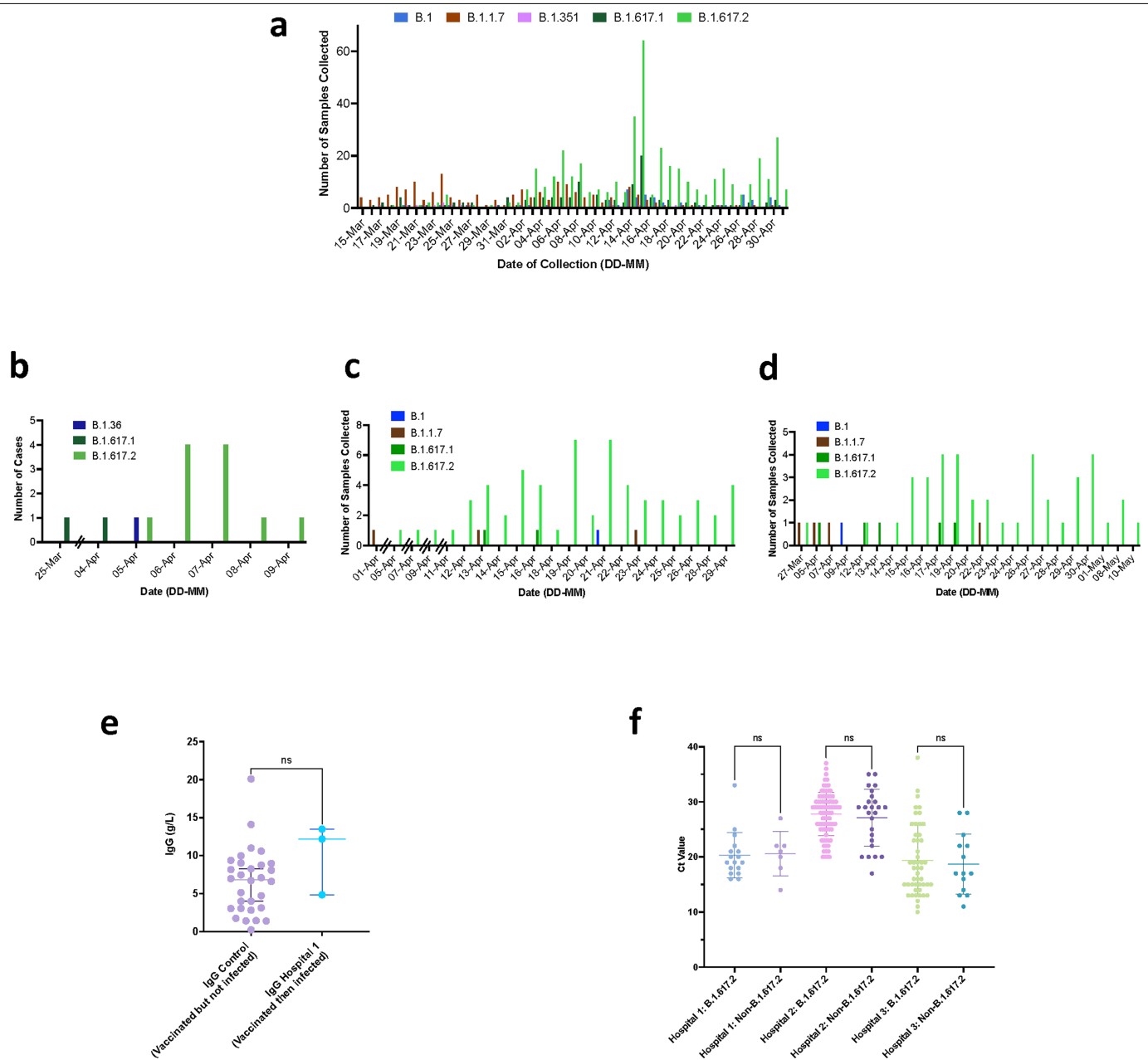

**Extended Data Fig. 4 | Breakthrough SARS-CoV-2 infections amongst vaccinated health care workers (HCW). a.** Case frequencies of five most commonly occurring SARS CoV-2 lineages over a six week period from March to April 2021 for Delhi **b,c,d**. case frequency graph for hospital 1, 2 and 3 respectively by date of testing. **e**. Comparison of IgG antibody titres between a control group of vaccinated individuals receiving two doses of ChadOx-1 who have not been infected with SARS-CoV-2, with vaccinated healthcare workers who had received two doses and subsequently tested positive for SARS-CoV-2. **f**. Ct values in nose/throat swabs from HCW testing positive by hospital. Bars represent Mean and 95% CI. Ct values were compared using the Student t test.

**Extended Data Table 1 | Demographic information for individuals undergoing two dose SARS-CoV-2 vaccination with ChAdOx-1 or BNT162b2**

| | ChAdOx-1 (N=33) | BNT162b2 (N=32) | P. value |
|---|---|---|---|
| Female (%) | 18 (54.5) | 13 (40.6) | 0.38[a] |
| Median Age *Years* (IQR) | 67 (64 - 71) | 71 (46 -83) | 0.74[b] |
| Median time *in Days* since dose 2 (IQR) | 31 (21 -38) | 27 (24 -29) | 0.15[b] |
| Serum Geometric Mean Titre *GMT* for delta variant (95% CI) | 654 (313 -1365) | 3372 (1856 - 6128) | 0.0006[b] |
| Serum Geometric Mean Titre *GMT* for WT (95% CI) | 2625 (1492 - 4618) | 7393 (3893 - 14041) | 0.0030[b] |

[a]Chi-square test, [b] Mann-Whitney test.

**Extended Data Table 2 | Monoclonal antibodies used in neutralisation assays against pseudotyped virus bearing spike from WT (Wuhan-1 D614) or B.1.617.2**

| mAb | Domain/site | IC50 WT (ng/ml) | IC50 B.1.617.2 (ng/ml) | VH usage (% Id.) | Source (DSD) | ACE2 blocking | Ref. |
|---|---|---|---|---|---|---|---|
| 82X107 | NTD | 2611.00 | 10000.00 | 4-38-2 (97) | Sympt. (75) | Neg. | McCallum et al. |
| 82X28 | NTD | 1121.30 | 10000.00 | 3-30 (97.9) | Svmpt. (48) | Neo. | McCallum et al. |
| 82X333 | NTD | 217.95 | 35016.00 | 3-33 (96.5) | Svmpt. (125) | Neo. | McCallum et al. |
| S2H7 | RBM | 1227.25 | 347.05 | 3-66 (98.3) | Svmpt. (17) | Weak | Thomson et al. |
| S2H14 | RBM | 1666.00 | 566.40 | 3-15(100) | Svmpt. (17) | Weak | Piccoli et al.: Thomson et al. |
| S2D19 | RBM | 369.55 | 129.95 | 4-31 (99.7) | Hosp. (49) | Moderate | Thomson et al. |
| 82X192 | RBM | 423.00 | 246.15 | 1-69 (96.9) | Svmpt. (75) | Weak | Thomson et al. |
| S2H19 | RBM | 549.25 | 382.45 | 3-15(98.6) | Svmpt. (45) | Weak | Thomson et al. |
| S2E12 | RBM | 2.09 | 1.72 | 1-58 (97.6) | Hosp. (51) | Strong | Thomson et al.; Tortorici et al. |
| 82X615 | RBM | 9.80 | 14.59 | 3-11 (94.8) | Sympt. (271) | Strong | Collier et al. |
| 82X128 | RBM | 36.82 | 65.97 | 1-69-2 (97.6) | Svmpt. (75) | Strong | Thomson et al. |
| S2D8 | RBM | 8.44 | 16.78 | 3-23 (96.5) | Hosp. (49) | Strong | Thomson et al. |
| S2H58 | RBM | 5.21 | 12.55 | 1-2 (97.9) | Svmpt. (45) | Strong | Thomson et al. |
| S2N28 | RBM | 10.05 | 36.95 | 3-30 (97.2) | Hosp. (51) | Strong | Thomson et al. |
| S2M11 | RBM | 2.07 | 8.32 | 1-2 (96.5) | Hosp. (46) | Weak | Thomson et al.: Tortorici et al. |
| S2D106 | RBM | 8.0 | 34.6 | 1-69 (97.2) | Hosp. (98) | Strong | Thomson et al. |
| S2D32 | RBM | 4.6 | 104.6 | 3-49 (98.3) | Hosp. (49) | Strong | Thomson et al. |
| S2N22 | RBM | 16.8 | 543.4 | 3-23 (96.5) | Hosp. (51) | Strong | Thomson et al. |
| S2D97 | RBM | 10.0 | 332.9 | 2-5 (96.9) | Hosp. (98) | Weak | Thomson et al. |
| S2H71 | RBM | 18.2 | 622.8 | 2-5 (99) | Svmpt. (45) | Moderate | Thomson et al. |
| S2H70 | RBM | 194.9 | 10000.0 | 1-2 (99) | Svmpt. (45) | Weak | Thomson et al. |
| 32X58 | RBM | 14.4 | 10000.0 | 1-46 (99) | Sympt. (48) | Strong | Thomson et al. |
| S2N12 | RBM | 10.6 | 10000.0 | 4-39 (97.6) | Hosp. (51) | Strong | Thomson et al. |
| 32X30 | RBM | 10.0 | 10000.0 | 1-69 (97.9) | Svmpt. (48) | Strong | Thomson et al. |
| etesevimab | RBM | 28.7 | 23.0/10.3* | 3-66 (99.7) | Svmpt. (?) | Strong | R. Shi et al. Nature 2020 |
| casirivimab | RBM | 6.0 | 7.2 / 3.5* | 3-30 (98.6) | Immunized | Strong | J. Hansen et al. Science 2020 |
| reqdanvimab | RBM | 2.2 | 29.2/16.3* | 2-70 (?) | Svmpt. (?) | Strong | |
| imdevimab | RBM | 31.9 | 1607.2 / 45.4* | 3-11 (98.6) | Svmpt. (?) | Strong | J. Hansen et al.Science 2020 |
| bamlanivimab | RBM | 7.8 | 10000/10000* | 1-69 (99.7) | Svmpt. (?) | Strong | Jones et al.Sci Transl Med 2021 |
| 3309 | non-RBM | 63.9 | 46.4 | 1-18(97.2) | SARS-CoV | Weak | Pinto etal. |
| 32X35 | non-RBM | 181.2 | 224.6 | 1-18(98.6) | Svmpt. (48) | Strong | Piccoli et al. |
| S2H94 | non-RBM | 134.1 | 182.9 | 3-23 (93.4) | Sympt. (81) | Strong | Thomson et al. |
| 82X259 | non-RBM | 74.2 | 101.1 | 1-69 (94.1) | Svmpt. (75) | Moderate | Tortorici et al (BioRxiv 2021) |
| S2H97 | non-RBM | 599.3 | 1260.4 | 5-51 (98.3) | Sympt. (81) | Weak | Collier et al.; Starr et al (BioRxiv 21) |
| 82X609 | non-RBM | 19.7 | 10000.0 | 1-69 (93.8) | Sympt. (271) | Strong | Collier et al. |
| 82X608 | non-RBM | 18.9 | 10000.0 | 1-33 (93.2) | Svmpt. (271) | Strong | Collier et al. |
| 82X619 | non-RBM | 18.2 | 10000.0 | 1-69 (92.7) | Svmot. (271) | Strong | Collier et al. |
| 82X305 | non-RBM | 11.8 | 9522.5 | 1-2 (95.1) | Svmpt. (125) | Strong | Collier et al. |

*in TMPRSS2 expressing VeroE6 cells.

**Extended Data Table 3 | Data on SARS-CoV-2 infections in three hospitals with near universal staff vaccination during first half of 2021**

| | B.1.617.2 (N= 112) | Non-B.1.617.2 (N=20) | P value |
|---|---|---|---|
| Median age *years* (IQR) | 36.5 (27.0-49.5) | 32.5 (27.5-44.0) | 0.56[a] |
| Female % | 51.8 (58) | 50.0 (10) | 0.88[b] |
| Hospital % <br> 1 <br> 2 <br> 3 | <br> 9.8 (11) <br> 53.6 (60) <br> 36.6 (41) | <br> 15.0 (3) <br> 30.0 (6) <br> 55.0 (11) | <br> 0.15[b] |
| Median Ct value (IQR) | 22.5 (16.4-28.6)[c] | 19.8 (17.3-22.8) | 0.48 |
| Number of vaccines doses %[†] <br> 0 <br> 1 <br> 2 | <br> 10.8 (12) <br> 20.7 (23) <br> 68.5 (76) | <br> 35.0 (7) <br> 30.0 (6) <br> 35.0 (7) | <br> 0.005[b] |
| Hospitalised %* <br> No <br> Yes | <br> 95.5 (64) <br> 4.5 (3) | <br> 93.3 (14) <br> 6.7 (1) | <br> 0.72[b] |
| Anti-Spike IgG GMT (95% CI) | 15.5 (4.6-52.9)[d] | 29.5 (0.0-2.4x10$^{6}$)[e] | 0.69[a] |
| Median Symptom duration *days* | 1.5 (1.0-3.0)[f] | 1.0 (1.0-2.0)[g] | 0.66[a] |

[a] Wilcoxon rank-sum test. [b]Chi square test. [c]111 of 112 available. [d]11 of 112. [e]2 of 20. [f]63 of 112. [g] 12 of 20. [†]Vaccine status missing for 1 of 132. *Hospitalisation data is unavailable from Hospital 1. IQR- inrerquartile range, GMT- geometric mean titre. CI- confidence interval.

**Extended Data Table 4 | Relative recei1 vaccine effectiveness against B1.617.2 v non- B1.617.2: Upper Table: Odds ratios for detection of B.1.617.2 relative to non-B.1.617.2 in vaccinated compared to unvaccinated individuals in multi-variable logistic regression**

| | B1.617.2 | Non-B.1.617.2 | B1.617.2: Non-B.1.617.2 | OR (95% CI) | P value | aOR (95% CI) | P value |
|---|---|---|---|---|---|---|---|
| Unvaccinated | 12 | 7 | 1.71 | - | | - | |
| | | | | | | | |
| Vaccinated | | | | | | | |
| Dose 1 | 23 | 6 | 3.83 | 2.24 (0.61-8.16) | 0.22 | 2.18 (0.53-9.01) | 0.28 |
| Dose 2 | 76 | 7 | 10.86 | 6.33 (1.89-21.27) | 0.003 | 5.45 (1.39-21.4) | 0.015 |
| Dose 1 and 2 | 99 | 13 | 7.62 | 4.44 (1.48-13.30) | 0.008 | 3.81 (1.11-13.03) | 0.03 |

| Model includes covariates | OR for B.1.617.2 vs non-B.1.617.2 (95% CI) | P value | OR for age (95% CI) | P value | OR for sex (95% CI) | P value | OR for hospital (95% CI) | P value |
|---|---|---|---|---|---|---|---|---|
| Dose 1 and 2 | 4.44 (1.48-13.30) | 0.008 | | | | | | |
| +age | 4.23 (1.34-13.31) | 0.014 | 1.01 (0.97-1.05) | 0.78 | | | | |
| +sex | 4.43 (1.48-13.29) | 0.008 | | | 0.96 (0.36-2.57) | 0.93 | | |
| +hospital<br>Hospital 1<br>Hospital 2<br>Hospital 3 | 4.64 (1.45-14.80) | 0.01 | | | | | Baseline<br>3.71 (0.77-17.94)<br>1.54 (0.34-6.96) | -<br>0.10<br>0.58 |
| +age +sex | 4.14 (1.29-13.28) | 0.017 | 1.01 (0.96-1.05) | 0.74 | 0.89 (0.31-2.60) | 0.84 | | |
| +age +sex +hospital<br>Hospital 1<br>Hospital 2<br>Hospital 3 | 3.81 (1.11- 13.03) | 0.03 | 1.03 (0.98-1.09) | 0.22 | 1.54 (0.48-4.97) | 0.47 | Baseline<br>8.69 (1.19-63.37)<br>2.27 (0.45-11.53) | -<br>0.03<br>0.32 |
| +age spline | 4.49 (1.38-14.48) | 0.011 | | | | | | |
| +week linear effect | 6.68 (1.71-27.40) | 0.006 | | | | | | |
| +week iid effect | 6.49 (1.6-28.8) | 0.009 | | | | | | |

Bottom table shows sensitivity of model to iterative addition of covariates. OR; odds ratio aOR; Adjusted odds ratio.

# Reporting Summary

Nature Research wishes to improve the reproducibility of the work that we publish. This form provides structure for consistency and transparency in reporting. For further information on Nature Research policies, see our Editorial Policies and the Editorial Policy Checklist.

## Statistics

For all statistical analyses, confirm that the following items are present in the figure legend, table legend, main text, or Methods section.

| n/a | Confirmed | |
|---|---|---|
| ☐ | ☒ | The exact sample size (*n*) for each experimental group/condition, given as a discrete number and unit of measurement |
| ☐ | ☒ | A statement on whether measurements were taken from distinct samples or whether the same sample was measured repeatedly |
| ☐ | ☒ | The statistical test(s) used AND whether they are one- or two-sided *Only common tests should be described solely by name; describe more complex techniques in the Methods section.* |
| ☐ | ☒ | A description of all covariates tested |
| ☒ | ☐ | A description of any assumptions or corrections, such as tests of normality and adjustment for multiple comparisons |
| ☐ | ☒ | A full description of the statistical parameters including central tendency (e.g. means) or other basic estimates (e.g. regression coefficient) AND variation (e.g. standard deviation) or associated estimates of uncertainty (e.g. confidence intervals) |
| ☐ | ☒ | For null hypothesis testing, the test statistic (e.g. $F$, $t$, $r$) with confidence intervals, effect sizes, degrees of freedom and $P$ value noted *Give P values as exact values whenever suitable.* |
| ☒ | ☐ | For Bayesian analysis, information on the choice of priors and Markov chain Monte Carlo settings |
| ☒ | ☐ | For hierarchical and complex designs, identification of the appropriate level for tests and full reporting of outcomes |
| ☐ | ☒ | Estimates of effect sizes (e.g. Cohen's *d*, Pearson's *r*), indicating how they were calculated |

*Our web collection on statistics for biologists contains articles on many of the points above.*

## Software and code

Policy information about availability of computer code

| Data collection | Graphad Prism v9.0.2 were used to produce figures. Mafft v7.475 was used for multiple sequence alignments. IQTREE and ModelFinder v2.1.4 was used to infer maximum-likelihood phylogenies. R v4.1.0 and ggplot package v3.3.3 were used to annotate phylogenies. |
|---|---|
| Data analysis | NextClade server v0.14.4 and Pangolin v3.0.5 were used to assign lineages to sequences. Pymol Graphics Suite v2.4.0 was used to visualize and annotate 3D protein structures Stata v13 was used for statistical analyses. |

For manuscripts utilizing custom algorithms or software that are central to the research but not yet described in published literature, software must be made available to editors and reviewers. We strongly encourage code deposition in a community repository (e.g. GitHub). See the Nature Research guidelines for submitting code & software for further information.

## Data

Policy information about availability of data

All manuscripts must include a data availability statement. This statement should provide the following information, where applicable:

- Accession codes, unique identifiers, or web links for publicly available datasets
- A list of figures that have associated raw data
- A description of any restrictions on data availability

Sequences from SARS-CoV-2 were obtained from GISAID database (https://gisaid.org/) using the filters and search parameters defined in the methods section. Structural models were obtained from the Protein Data Bank (PDB) https://www.rcsb.org/.
All fasta consensus sequences files donated by collaborators are freely available from Gisaid (https://gisaid.org) with accession numbers as follows: Hospital 1:

# Field-specific reporting

Please select the one below that is the best fit for your research. If you are not sure, read the appropriate sections before making your selection.

☒ Life sciences  ☐ Behavioural & social sciences  ☐ Ecological, evolutionary & environmental sciences

For a reference copy of the document with all sections, see nature.com/documents/nr-reporting-summary-flat.pdf

# Life sciences study design

All studies must disclose on these points even when the disclosure is negative.

| | |
|---|---|
| Sample size | This is a descriptive study of an outbreak of SARS-CoV-2 in 3 hospitals and so a sample size is not appropriate. |
| Data exclusions | For sequencing data, consensus fasta files were excluded, as described in the methods section, based on poor coverage of the genomes. |
| Replication | Experiments were done in technical duplicates and each experiment was repeated. |
| Randomization | Not applicable as this is not an intervention study. |
| Blinding | Patients were pseudoanonymised at point of receiving all data from collaborators. All reported data in phylogenies are blinded and only show values and lineage assignments. |

# Reporting for specific materials, systems and methods

We require information from authors about some types of materials, experimental systems and methods used in many studies. Here, indicate whether each material, system or method listed is relevant to your study. If you are not sure if a list item applies to your research, read the appropriate section before selecting a response.

## Materials & experimental systems

| n/a | Involved in the study |
|---|---|
| ☐ | ☒ Antibodies |
| ☐ | ☒ Eukaryotic cell lines |
| ☒ | ☐ Palaeontology and archaeology |
| ☒ | ☐ Animals and other organisms |
| ☐ | ☒ Human research participants |
| ☒ | ☐ Clinical data |
| ☒ | ☐ Dual use research of concern |

## Methods

| n/a | Involved in the study |
|---|---|
| ☒ | ☐ ChIP-seq |
| ☒ | ☐ Flow cytometry |
| ☒ | ☐ MRI-based neuroimaging |

# Antibodies

| | |
|---|---|
| Antibodies used | anti-SARS-CoV-2 Spike (Invitrogen, PA1-41165, 1:5,000)<br>anti-GAPDH (Proteintech,60004, 1:5,000)<br>anti-p24 (NIBSC, ARP365 and ARP366, 1:500; ARP313, 1in10,000)<br>anti-rabbit HRP (Cell Signaling, 7074S, 1:3,000)<br>anti-b-actin HRP (Abcam, ab8226, 1:10, 000) |
| Validation | Validation was conducted by manufacturers prior to sale. No further validation was undertaken. |

# Eukaryotic cell lines

Policy information about cell lines

| | |
|---|---|
| Cell line source(s) | HeLa ACE2 cells were donated by kind request from James Voss as noted in the methods section.<br>HEK 293T CRL-3216 cells from ATCC were used for transfection work.<br>H1299 cells were a gift from Simon Cook as noted in methods.<br>Calu-3 cells were a gift from Paul Lehner as mentioned.<br>A549 ACE2T/TMPRSS2 cells were a kind gift from Massimo Palmerini.<br>Vero E6 ACE2/TMPRSS2 cells were a gift from Emma Thomson.<br>HAE cells (MucilAir™) were purchased from Epithelix. |

293T cells were purchased from Takara Bio (# 632180)
Airway epithelial organoids were prepared and donated by Joo-Hyeon Lee as described in (10.1016/j.stem.2020.10.004)

| | |
|---|---|
| Authentication | None of the cell lines used were authenticated. |
| Mycoplasma contamination | All cell lines used were tested (by PCR) and were mycoplasma free. |
| Commonly misidentified lines (See ICLAC register) | No commonly misidentified lines were used in this study. |

# Human research participants

Policy information about studies involving human research participants

| | |
|---|---|
| Population characteristics | Participants include health care workers involved in an outbreak of SARS-CoV-2 in 3 hospitals in India. Vaccine sera were obtained from participants involved. |
| Recruitment | As part of routine testing, venous serum samples were collected from the participants enrolled in the NIHR BioResource Centre Cambridge |
| Ethics oversight | Ethical approval for use of serum samples. Controls with COVID-19 were enrolled to the NIHR BioResource Centre Cambridge under ethics review board (17/EE/0025). Convalescent sera from healthcare workers at St. Marys Hospital at least 21 days since PCR678 confirmed SARS-CoV-2 infection were collected in May 2020 as part of the REACT2 study with ethical approval from South Central Berkshire B Research Ethics Committee (REC ref: 680 20/SC/0206; IRAS 283805). Studies involving health care workers (including testing and sequencing of respiratory samples) were reviewed and approved by The Institutional Human Ethics Committees of NCDC and CSIR-IGIB(NCDC/2020/NERC/14 and CSIR-IGIB/IHEC/2020-21/01). All participants provided informed consent. |

Note that full information on the approval of the study protocol must also be provided in the manuscript.

