## [Peer Review File · Nature]

Manuscript Title: SARS-CoV-2 B.1.617.2 Delta variant replication and immune evasion

Reviewer Comments & Author Rebuttals

Reviewer Reports on the Initial Version:

Referees' comments:

Referee #1 (Remarks to the Author):

In their manuscript Mlcochova and colleagues look into four aspects of B.1.617.2. They describe infections and spread of the virus in HCWs fully vaccinated with the AZ vaccine. That part is novel and interesting. They do some characterization of the susceptibility of the variant to virus neutralization. This part is limited by very small sample numbers, different assays used and the fact that a lot of neutralization data for B.1.617.2 has already been published. They also characterize the enhanced cleavage and syncytia formation of B.1.617.2 which is novel and interesting. Finally, they try to model several aspects of B.1.617.2 epidemiology. This part is mere speculation partially informed by wrong assumptions.

Major points

- 1) The title should be changed to 'and ChAdOx1 nCoV-19 vaccine breakthrough'. No other vaccines are evaluated and the effectiveness of the mRNA vaccines against B.1.617.2 is good. Generalization is unwarranted, the authors should use a title based on the data. This also needs to be revised throughout the manuscript. Also, the Greek names are for the lay press. Please use a scientific lineage name.
- 2) The manuscript has no page or line numbers.
- 3) The authors highlight immune evasion throughout the manuscript as a driving force for the spread of B.1.617.2 but have no hard evidence for that at all. They describe that 'background infection' in India during the first quarter of 2021 was 20-50%. This assumption is plainly wrong. First, it is known by know that the specificity of serology tests differs depending on where they are used. The specificity is much lower in some African countries as compared to e.g. the US. The same is true for India. These estimates of 20%-50% infection rates are plainly based on flawed data (like in Brazil). Also, by Q1 2021, 12 million official cases had been reported in India and 164 000 official deaths, resulting in a CFR of about 1.36%. Sure, there was likely underreporting for both. But assuming 20-50% of the population were infected by that time, the CFR would be between 0.06% and 0.024% which makes no sense. Even if deaths were underreported by a factor of three (which is realistic) this still makes no sense. Using these numbers to inform a model leads to a wrong model. Any language about the role of reinfection for driving B.1.617.2 spread should be removed unless the rate of reinfection can be measured (not predicted, measured – hard data!) based on large numbers.
- 4) Differences in replication are measured by genome copy number. Plaque forming units need to be added for all experiments. Infectious virus is what matters, genome copy numbers can be way off from infectious units.
- 5) Serology was only performed on a very small number of samples. This should be extended.
- 6) A variety of different assays for neutralization were used. Are they comparable? It would have been better to measure everything in one type of assay.
- 7) On page 6 the authors state that there were no clusters with more than 2 individuals with non-Delta variants. The extended data figure 1D says otherwise.

Minor points

- 1) Throughout the manuscript: Many abbreviations are not defined.
- 2) Page 3, first line: Remove 'has'.
- 3) Page 3: The California lineage consists of two lineages.

- 4) The discussion section is full of mistakes including word duplications and sentences that end mid-sentence etc.
- 5) Page 9: B.1.617.1 no, B1.1.617.1
- 6) Page 11: What is meant by 'COVIVAX vaccinates'?
- 7) Page 15: SARS-CoV-2 is misspelled.
- 8) Figure 1C: It is not clear what is shown here.
- 9) Figure 1D: The underlying assumptions for this are wrong.
- 10) Figure 2B: Please include PFUs for all measures.
- 11) Figure 2D: Please add positive and negative controls for the plots.
- 12) Figure 4A: The variant names on the X-axis are misspelled.
- 13) Extended Figure 2C: Add positive and negative controls.

Referee #2 (Remarks to the Author):

Overall, this reviewer's view is that this manuscript represents a highly relevant study, with significant current interest, with the work well done. A primary concern is that the key infectivity findings refer to B.1.617 in general, and not B.1.617.2 ("Delta") specifically, and so this should be made fully clear in the manuscript, with the caveat that these data do not really explain the current prevalence (or fitness advantage) of B.1.617.2 over B.1.617.1 (for instance).

- 1) The Introduction begins in a very vague way, and should be improved to focus the reader on SARS-CoV-1/COVID-19. The "California" variant of the only one described in this way; please update. In general, the Introduction is very brief.
- 2) While the use of primary organoids is an excellent technique, the replication data with infectious virus should be complemented with studies in cell lines that are more available, e.g. Calu3 compared to VeroE6/Vero-TMPRSS2 – if nothing else in order to compare data here with other studies that will almost certainly come on this variant
- 3) More complete data needs to be presented on virus infectivity, beyond the qPCR data presented – to include TCID50/plaque assay
- 4) Is there a significant difference between B.1.617.2 and B.1.617.1 with camostat treatment? The IC50 would appear to indicate this is the case. In any event Fig 2f should be expanded and repeated with infectious virus, as it does seem to be a significant difference
- 5) Statistical analysis is also missing from Figure 1f
- 6) For a scientific paper, this reviewer considers that terminology should be standardized on B.1.1.617.2 (and not delta, alpha etc)
- 7) Is "WT" the correct terminology? (Figure 2) - this is better as B.1 (or Wuhan-1 or whatever is most appropriate)

Line numbers and page numbers should be used in the manuscript for an optimal review process

Referee #3 (Remarks to the Author):

Mlcochova present a series of findings on the COVID Delta variant, including some epidemiological observations and some in-vitro observations. My comments are focused on the epidemiological observations.

Major comments

1. No methodology is provided for the modeling analysis which forms the motivation for the remainder of the paper. Even without a modeling analysis it is clear from the rapid growth in Delta variant outside of India that it is more transmissible than previous variants. Authors use a model here to infer that Delta is also overcoming natural immunity from prior non-Delta infections to a greater extent than other strains. Without details of the model it is not possible to appraise it. Authors included "waning of immune protection following infection, parameterised using the results of recent longitudinal cohort studies" - was this waning in antibodies or waning in protection against confirmed (re-)infection? How are you accounting in the model for heterogeneity in exposure, i.e. people such as HCWs and workers in service sector more likely to be exposed to infection in 2020 are also more likely to be exposed to infection in 2021?

2. In the section "Transmission clusters in vaccinated health care workers", authors hypothesised that antigenic changes were responsible for infections in healthcare workers, but did not estimate vaccine effectiveness or demonstrate that vaccine effectiveness had changed for the Delta variant. In Extended Data Figure 1 panels c-e the cluster sizes are larger with Delta lineage but it is not clear whether sampling was systematic, and whether a small number of larger clusters might have occurred by chance. Authors did find significant differences in Ct values and please confirm that this comparison was adjusted for time since symptom onset.

3. Regarding the analysis of cluster sizes, in epidemiology usually a cluster is defined as 2 or more linked cases, so I'm not sure how a mean cluster size can be less than 2. If you mean statistical clusters of the same virus sequence, probably you need to think about the underlying structure of the outbreaks and think about how to model un-sequenced infections. Many healthcare workers could be infected in the community by family members, or by their own (possibly different) patients, rather than each other, meaning that large clusters of infections in HCWs may not be that common. Given that you did not attempt to identify all infections with for example mass testing, nor did you sequence all confirmed cases, a more complex analysis may be required.

Minor comments

Introduction "The B.1.617 variant emerged in the state of Maharashtra in late 2020/early 2021" - are you confident it emerged here, or just that it was first detected here?

Referee #4 (Remarks to the Author):

- Summary

In this study, Mlcochova and colleagues report of the emergence of the Delta variant of SARS-CoV-2 in India and present data on Delta kinetics, immune evasion, vaccine breakthroughs, and syncytium formation. The study contains some valuable study, but the majority of the conclusions are currently not supported by the analyses performed by the authors. There is a great need to much more carefully perform the majority of the analyses in the manuscript and generally speaking focus much more on making conclusions robust and representative. For these reasons I cannot recommend publication at this stage and a major revision will be required.

- Major points:

Their "Bayesian model of SARS-CoV-2 transmission" needs to be much better justified and additional analyses are required to establish robustness. Did the authors perform any sensitivity analyses? How dependent are they on choice of priors? How exactly did they run the model? No details have been provided in the methods.

The experiments in Fig. 2a,b appear to be very preliminary. The detected amounts are extremely low (e.g. 10^{-2} / cell or 10 copies / ml) and therefore very sensitive to experimental variability. These experiments should be repeated and proper statistics applied. How are these experiments normalized to control for slight differences in measured MOIs? For the remaining experiments in this figure, why not compare to B.1.1.7? How does Delta differ compared to Alpha?

"We found that B.1.617.2 was marginally less sensitive to the TMPRSS2 inhibitor Camostat" - proper stats need to be

applied here and a statistical statement made.

The whole "Transmission clusters in vaccinated health care workers associated with Delta variant" needs to be reanalyzed. Sequences are not independent and it's impossible to distinguish transmission within a cluster vs those sourced from the community by using criteria such as "we defined related or 'linked' infections as differing by six nucleotides or less". This is especially true since B.1.617.2 is overall more diverse than e.g., B.1.1.7 and has a different history, hence conclusions about "mean cluster size" end up being fraught with problems. This whole section will require significant additional analyses and a rewrite - it's very unclear what exactly the authors did and what can be concluded based on the data and analyses. I don't doubt that the authors may be correct in some of their conclusions, but the current set of data and analyses are insufficient to support the conclusions made in this section. In the abstract it's also mentioned that "Delta variant [...] dominates vaccine-breakthrough infections", but it's unclear if this is simply due to its prevalence or if it's because it's somehow better able than e.g., alpha in causing vaccine breakthroughs. That is not clear from the present study.

"The B.617.1 and B.617.2 spikes demonstrated higher fusion activity and syncytium formation, mediated specifically by P681R" The authors have previously done these experiments for alpha - how do they compare?

All data should be made available immediately on GISAID. The authors use publicly available data (via GISAID), but have not themselves made their data available - that is not acceptable.

The entire discussion section needs work. It's currently very confusing, meanders through different findings, and contains several half-sentences, missing references, and run-on sentences.

- Minor points:

Typos and inconsistencies throughout. E.g., from abstract "ChAdOx-1" should either be "ChAdOx1" or "AZD1222"; "HCW", "GMT" are undefined; "shows 8 fold approximately reduced sensitivity" should be "shows approximately 8-fold reduced sensitivity", etc.

Introduction about the epidemic in India should be updated to reflect current situation and careful attention should be given to properly introducing the background and questions - it's a bit of a mess at the moment. Referencing all needs to be improved and the usage of "California" variant should be avoided (the WHO has designated this variant as "Epsilon").

Author Rebuttals to Initial Comments:

Referee #1

(Remarks to the Author)

In their manuscript Mlcochova and colleagues look into four aspects of B.1.617.2. They describe infections and spread of the virus in HCWs fully vaccinated with the AZ vaccine. That part is novel and interesting. They do some characterization of the susceptibility of the variant to virus neutralization. This part is limited by very small sample numbers, different assays used and the fact that a lot of neutralization data for B.1.617.2 has already been published. They also characterize the enhanced cleavage and syncytia formation of B.1.617.2 which is novel and interesting. Finally, they try to model several aspects of B.1.617.2 epidemiology. This part is mere speculation partially informed by wrong assumptions.

Response: We thank the reviewer for this comment. We have taken additional time to construct a comprehensive, multi-dimensional piece of work rather than publishing limited datasets and single figure articles as some other groups have done recently. We hope that

this approach can be appreciated. We have added significant new infection and monoclonal antibody neutralisation data that is novel and important.

Major points

- 1) The title should be changed to 'and ChAdOx1 nCoV-19 vaccine breakthrough'. No other vaccines are evaluated and the effectiveness of the mRNA vaccines against B.1.617.2 is good. Generalization is unwarranted, the authors should use a title based on the data. This also needs to be revised throughout the manuscript. Also, the Greek names are for the lay press. Please use a scientific lineage name.

Response: we have considered titles and agree about being specific if we refer to vaccines. However we are happy to defer to the editors about choice of title, including a more general title such as 'SARS-CoV-2 B.1.617.2 emergence, replication and sensitivity to neutralising antibodies'

- 2) The manuscript has no page or line numbers.

Response: These have now been inserted, we apologise for this omission

- 3) The authors highlight immune evasion throughout the manuscript as a driving force for the spread of B.1.617.2 but have no hard evidence for that at all. They describe that 'background infection' in India during the first quarter of 2021 was 20-50%. This assumption is plainly wrong. First, it is known by know that the specificity of serology tests differs depending on where they are used. The specificity is much lower in some African countries as compared to e.g. the US. The same is true for India. These estimates of 20%-50% infection rates are plainly based on flawed data (like in Brazil). Also, by Q1 2021, 12 million official cases had been reported in India and 164 000 official deaths, resulting in a CFR of about 1.36%. Sure, there was likely underreporting for both. But assuming 20-50% of the population were infected by that time, the CFR would be between 0.06% and 0.024% which makes no sense.

Even if deaths were underreported by a factor of three (which is realistic) this still makes no sense. Using these numbers to inform a model leads to a wrong model. Any language about the role of reinfection for driving B.1.617.2 spread should be

removed unless the rate of reinfection can be measured (not predicted, measured – hard data!) based on large numbers.

Response: We thank the reviewer for their comment. While there are a number of serosurveys for India (and Brazil) which have obtained differing results, we strongly disagree with the dismissal of the possibility of high attack rates. Regarding the claim about specificity, there is sufficient data from multiple validated instruments showing that estimates of 20-50% seropositivity in India in Q1 2021 are real. In the CSIR cohort serosurvey, over 10,000 employees have been tested over 17 states and 2 union territories in two completed phases, namely Aug-Sept 2020, Feb-Mar 2021, and an ongoing phase. The primary instrument to determine seropositivity was an Electro-chemiluminescence Immunoassay (ECLIA)-Elecsys Anti-SARS-CoV-2 kit (Roche Diagnostics) measuring antibodies to SARS-CoV-2 NC antigen. This assay has been considered a method of choice when a single test is to be deployed (Krüttgen et al., 2021). Positive results were further tested for neutralizing antibody (NAB) response directed against the spike protein using GENScript cPass SARS-CoV-2 Neutralization Antibody Detection Kit (GenScript, USA). In 10,427 subjects tested in Phase I, 1058 were seropositive for Anti-SARS-CoV-2 NC-antigen. Of these, 1025 subjects were further tested for neutralization, with 976 being positive on the NAB test (95.2% concordance). The results of the serosurvey and the source data-files have been peer-reviewed and published (Naushin et al, eLife 2021;10:e66537 DOI: [10.7554/eLife.66537](https://doi.org/10.7554/eLife.66537) ; source data file from the publication is available at <https://cdn.elifesciences.org/articles/66537/elifesciences-66537-fig1-data1-v2.docx>). In additional experiments for assessment of concordance between NC and Spike assays, in 201 samples positive for antibodies against nucleocapsid antigen (N-ab), we also measured Anti-spike (Roche Diagnostics), finding 193 (97%) to be positive. Together, this excludes the possibility that results of the CSIR serosurvey are flawed due to lack of specificity of the anti-NC test in India.

Seropositivity rose in this pan-India cohort from 10.14% in Q3 2020 (Phase I) to 29% in Q1 2021 (Phase 2). During Phase I, when the pandemic was yet to fully spread, seropositivity was as low as 1% in some cities, while it was over 10% in others, consistent with known epidemiologic indicators (Naushin et al, eLife. 2021;10:e66537). This excludes any concerns of background false positivity in India that would distort the results of the serosurvey. In Phase 2, using the same instruments

in the same population, seropositivity rates in most cities were over 40% while smallertowns were around 20%. For example Delhi, with 465 of 1115 subjects positive, had about 42% seropositivity. These samples were again further tested for Anti-Spike- RBD (Roche) and NAB as earlier, with 97.7% and 88.7% being positive respectively. This represented a sharp rise from 14% in Q3 2020 and emphatically confirms very high attack rates in Delhi, falling within the ranges used in the model for Mumbai, a comparable Megacity. See preprint (<https://www.medrxiv.org/content/10.1101/2021.06.02.21258076v1>, manuscript in review). It is noted that these crude rates are without accounting for factors such as exposures, seroconversion and seroreversion rates, which would probably lead to further upward revisions since CSIR labs do not fully represent high-exposure settings such as slums. Seroconversion occurs in about 85% of confirmed cases from similar settings (see Ramachandran et al, Am J Trop Med Hyg. 2021 May 18;tpmd210164), and sero-reversion rates were about 10% over this period. Thus, the range of 20-50% attack rate for Q1 2021 for Mumbai is grounded in valid data. It is further noted that sero-surveys are useful for calculating infection fatality rate (IFR), not case fatality rate (CFR). Calculated IFRs based on CSIR survey as well as other national surveys in India have been between 0.05 and 0.1%. These are plausible, given the range of death underreporting accounted for in the model.

The above results highlight that the specificity of the serological tests used are unlikely to be an issue that significantly alters the level of exposure inferred in the population, at least not to degrees that influence the qualitative conclusions presented here that some degree of both transmissibility increase and immune-evasion must be invoked to explain the patterns observed in Mumbai. An additional important point however is that because the serological estimates included in the modelling framework used here do not correct for seroreversion, they are likely to underestimate the degree of exposure in the population. We do however recognise that there are variety of different factors that might bias these serological results, and to that end, the modelling framework explicitly incorporates sampling variation in the serological results as part of the observational model used to fit to the data.

In addition to the above comments on the validity of the serological data used in the modelling framework, we wish to provide more technical detail about the modelling

framework itself. Whilst the model is broadly described in the Supplementary Information of Faria et al, (Science, 2021) we apologise for the absence of a technical appendix in the first submission of this manuscript. To facilitate full examination and amore comprehensive understanding of the model as used here, we enclose a technical appendix fully describing the model with this submission. We would welcome the reviewer's comments and any critical feedback on the structure and assumptions of themodel.

In addition to the model description, we include a simulation study to address the capacity of the model to correctly infer an absence of immune-evasion. We generate a synthetic dataset from a hypothetical setting in which a variant emerges that possesses no immune-evasion potential and is only more transmissible. We then fit our modellingframework to this data and explore the inferred degree of immune-evasion by the model. Our model correctly recovers the ground truth of no immune evasion, demonstrating that our framework correctly excludes the hypothesis of immune evasionwhen it is not borne out by the data.

We briefly summarise key aspects of the model, and further comments in response tothe reviewer's feedback, below.

A substantial body of literature exists developing the conceptual and modelling frameworks for understanding the dynamics and properties of pathogens that consist ofmultiple antigenic variants with limited degrees of immune cross-protection between variants (e.g. Kucharski et al, Journal of Mathematical Biology, 2016). The modelling results presented here are derived from an application of this multi-strain framework to a well-established class of semi-mechanistic, renewal equation-based infectious disease models. It has previously been used to quantify and characterise the dynamics and properties of the Gamma variant during its establishment in Manaus, Brazil (Fariaet al., Science 2021).

We respectfully disagree with the reviewer that the absence of "hard data" on reinfection rates means the presented framework is unsuitable for exploring the degree of immune evasion likely to be associated with B.1.617.2 spread. Previous work has defined confirmedreinfection as requiring confirmation of a true first episode (with PCR based detection of

infection) followed by PCR-based diagnosis of a second infection and confirmation of infection with two different phylogenetic strains (i.e. proof of two distinct virus variants with any sequence variation between the two episodes, to preclude the possibility of residual viral circulation in the body followed by recrudescence) using high-throughput sequencing approaches (Yahav et al., Clinical Microbiology and Infection, 2020). Given the testing and sequencing limitations India has faced, particularly during the first wave, generating conclusive “hard data” on reinfections would be incredibly difficult, if not impossible.

Moreover, even with this data, it would not be possible to disentangle the competing hypotheses of immune-evasion and natural waning of immunity in leading to these reinfections. Use of modelling allows us to disentangle these competing hypotheses and assess their comparative importance in the context of the dynamics observed in Mumbai. This is made possible by our framework’s explicit inclusion of waning of immunity, parameterised using results from recent published literature, including the SIREN Cohort Study in the United Kingdom (Hall et al, Lancet, 2021) and a large-scale observational study from Denmark (Holm Hansen et al, Lancet, 2021). Importantly, previous estimates about the degree of expected reinfection for results from the model produced during work characterising the Gamma variant (Faria et al, Science, 2021) were subsequently shown to be consistent with the pattern of reinfection observed from a longitudinally sampled blood donor cohort from the same city (that was not available at the time of the original study, Prete et al., medRxiv, 2021), highlighting the capacity of this framework to produce results later shown to be consistent with detailed clinical investigations.

We wish however to also emphasise that the above should not be interpreted as our saying that data on reinfection is not important – if available, it would represent a crucial input to the model that would further refine and reduce the uncertainty associated with the estimates presented here; and should be a priority for future studies. More generally, we also recognise the extensive uncertainty in our estimates regarding the exact comparative contributions of Delta’s increased transmissibility vs immune-evasion to the observed dynamics in Mumbai. We have therefore added additional text in the manuscript: (i) we caveat our conclusions in more uncertain terms; (ii) we present results from the model under the assumptions of either no transmissibility increase or no immune-evasion as these help to understand the posterior we infer; and (iii) we add in a number of caveats surrounding the data sources we use.

Finally, we would also note that the results presented here sit alongside an increasingly substantial body of evidence highlighting the propensity for Delta to (partially) evade immunity derived from vaccination (Stowe et al, Public Health England Knowledge Hub, 2021), and documented instances of Delta reinfections occurring in the United Kingdom (311as of Public Health England Technical Report 16, 2021). Recent work has highlighted that antibodies elicited by vaccination are broader in their antigenic targets than those elicited by SARS-CoV-2 infection, rendering vaccine-derived immunity less likely to be eroded by the antigenic variation characterising the Delta variant (Greaney et al, Science Translational Medicine, 2021). The observation of some degree of immune-evasion in vaccinated individuals is therefore consistent with a moderate degree of immune-evasion (to predominantly natural immunity, in the case of the modelling, given the relatively low levels of population-level vaccination coverage in India over the period considered), as described here. We have added a number of clarifying sentences to emphasise this point and provide additional context to readers surrounding previous research on the subject.

- 4) Differences in replication are measured by genome copy number. Plaque forming units need to be added for all experiments. Infectious virus is what matters, genome copy numbers can be way off from infectious units.

Response: It was not possible to do plaque assays in the organoid system but we have done TCID50 measurements in Calu3 as well as a range of experiments with read outs that included protein expression. We have also added data that include genome copy number as well as PFU from the HAE system that has an air liquid interface. All show that B.1.617.2 has significant replication advantage over B.1.1.7.

- 5) Serology was only performed on a very small number of samples. This should be extended.

Response: We do not have pre infection samples from greater numbers of vaccinees unfortunately. We would be happy to remove the data if the reviewer felt this was needed.

- 6) A variety of different assays for neutralization were used. Are they comparable? It would have been better to measure everything in one type of assay.

Response: we have not performed direct comparisons between assays and convalescent plasma assays were done in a different lab to the vaccine sera – given the pandemic this was necessary so as not to delay reporting. We feel that the important aspect is that the B.1.617.2 shows greater immune evasion than B.1.1.7 across all assays used. This is most pertinent as the whole premise of the work is why B.1.617.2 is now becoming the dominant variant across the globe, taking the place of B.1.1.7. We agree ideally that one would conduct all experiments with one system, though consistent findings across assays is in fact rather reassuring.

7) On page 6 the authors state that there were no clusters with more than 2 individuals with non-Delta variants. The extended data figure 1D says otherwise.

Response: we thank the reviewer for pointing out this inconsistency. Following peer review of the clustering analysis and additional work needed to confirm linked infections we have elected to remove this aspect given timeliness, journal space considerations, as well as importance of other aspects of our findings. The cluster analysis will form the basis for a separate report. Instead we have added an analysis of vaccine efficacy as suggested by reviewer 3 that is not dependent on linkage or clustering of infections. This analysis shows that B.1.617.2 is more likely to be isolated in breakthrough infections than non- B.1.617.2 viruses in the context of ChAdOx-1 vaccinated HCW, and this links with the findings from the rest of the manuscript.

Minor points

1) Throughout the manuscript: Many abbreviations are not defined.

Response: we have gone through the paper to expand abbreviations.

2) Page 3, first line: Remove 'has'.

Response: we have removed this

3) Page 3: The California lineage consists of two lineages.

Response: thank you for pointing this out, we have now clarified this.

- 4) The discussion section is full of mistakes including word duplications and sentences that end mid-sentence etc.

Response: we have now amended and checked the discussion and introduction.

- 5) Page 9: B.1.617.1 no, B1.1.617.1

Response: thank you we have corrected this

- 6) Page 11: What is meant by ‘COVIVAX vaccinates’?

Response: this should be ChAdOx-1 vaccinees.

- 7) Page 15: SARS-CoV-2 is misspelled.

Response: thank you we have corrected this

- 8) Figure 1C: It is not clear what is shown here.

Response: this is the reproductive number for B.1.617.2 v non- B.1.617.2 (stated in legend)

- 9) Figure 1D: The underlying assumptions for this are wrong.

Response: please see response above, we disagree with this comment as we have strong validated data for seropositivity rates. We do appreciate that early in the epidemic there was widespread use of tests that were unvalidated with poor specificity, but we can assure the reviewer that the data used to inform our models is from validated, highly specific serology tests.

- 10) Figure 2B: Please include PFUs for all measures.

Response: please see above, we have TCID50 for Calu-3 and HAE systems in the revised paper as well as protein expression by western blotting.

11) Figure 2D: Please add positive and negative controls for the plots.

Response: We have added negative controls for the infectivity plots though there is no positive control as such. The control in this experiment is the actin loading and probing for HIV-1 Gag in the PV.

12) Figure 4A: The variant names on the X-axis are misspelled.

Response: thank you we have corrected this to state B.1.1.7

13) Extended Figure 2C: Add positive and negative controls.

Response: we have performed blots with negative controls (no spike PV). We are interested in cleavage status of spike so there is no real positive control, as we are comparing a Wuhan-1 D614G Spike with the B.1.617.2 spike and looking at differences between them.

Referee #2

(Remarks to the Author)

Overall, this reviewer's view is that this manuscript represents a highly relevant study, with significant current interest, with the work well done. A primary concern is that the key infectivity findings refer to B.1.617 in general, and not B.1.617.2 ("Delta") specifically, and so this should be made fully clear in the manuscript, with the caveat that these data do not really explain the current prevalence (or fitness advantage) of B.1.617.2 over B.1.617.1 (for instance).

Response: we thank the reviewer for recognising the collaborative efforts across continents in this work. We would like to clarify that all the live virus work – infectivity, vaccine sera and convalescent sera has been done with a B.1.617.2 (Delta) live isolate, and comparisons made with B.1.1.7 in order to demonstrate that it has both immune evasion and fitness

advantages over B.1.1.7 that was also prevalent in India at the start of 2021, as shown below and in the new figure 3.

Above: HAE infection data (see figure 3)

Above: Calu-3 data – top intracellular RNA and protein, bottom extracellular RNA and TCID50 (see figure 3).

For some of the PV work we used a B.1.617.1 spike to compare with B.1.617.2 (Delta) spike. We agree that we cannot conclusively explain the current prevalence (or fitness advantage) of B.1.617.2 over B.1.617.1, though one can see in the PV entry experiments that B.1.617.2 spike had an advantage over B.1.617.1 spike and that the live B.1.617.2 virus has faster replication kinetics than the B.1.617.1 in Calu-3 lung cells. Although this is a secondary message in the paper it is nonetheless an important finding. The new live virus data are

shown below for B.1.617.1 v B.1.617.2 in Calu-3 lung cells.

- 1) The Introduction begins in a very vague way, and should be improved to focus the reader on SARS-CoV-1/COVID-19. The “California” variant of the only one described in this way; please update. In general, the Introduction is very brief.

Response: we have now elaborated the introduction with greater focus. We believe the reviewer meant SARS-CoV-2 rather than 1 here.

- 2) While the use of primary organoids is an excellent technique, the replication data with infectious virus should be complimented with studies in cell lines that are more available, e.g. Calu3 compared to VeroE6/Vero-TMPRSS2 – if nothing else in order to compare data here with other studies that will almost certainly come on this variant.

Response: we thank the reviewer for this comment and have now presented experiments to address this in Calu3 as well as in HAE systems. Data from all three model systems are consistent with a replication advantage for B.1.617.2.

- 3) More complete data needs to be presented on virus infectivity, beyond the qPCR data presented – to include TCID50/plaque assay

Response: we are not able to do plaque assay on the organoids, but we have taken the supernatants from the organoids and shown that there is infectious virus present at higher levels for the Delta variant. We have also performed western blotting of Calu-3 cells and supernatants to demonstrate that the qPCR correlates with viral protein production. We have done TCID50 in the Calu-3 experiments, and TCID50/plaque assay data are also shown for the additional HAE experiments in figure 3.

Above: a-d. Infection in Calu-3, a-b cell lysate analysis, c-d supernatant analysis

4) Is there a significant difference between B.1.617.2 and B.1.617.1 with camostat treatment?

The IC50 would appear to indicate this is the case. In any event Fig 2f should be expanded and repeated with infectious virus, as it does seem to be a significant difference

Response: When we performed the experiment with live virus the same result was not obtained. As this will require more detailed investigation we have removed the camostat data from this manuscript given the already broad scope of the paper.

5) Statistical analysis is also missing from Figure 1f

Response: the fold change with 95% CI shows it is significant for B.1.617.2 as 95%CI does not cross include 1. Given the conflicting results from PV and live virus, that may be due to effects of camostat on the virus life cycle, we need to spend more time investigating this aspect and have removed the panel from this manuscript.

6) For a scientific paper, this reviewer considers that terminology should be standardized on B.1.1.617.2 (and not delta, alpha etc)

Response: we have done this and mention in the introduction the relation to WHO nomenclature. Nature has a general science audience so we feel it important to mention the WHO classification somewhere.

7) Is “WT” the correct terminology? (Figure 2) - this is better as B.1 (or Wuhan-1 or whatever is most appropriate)

Response: we have clarified this in the legend thank you.

Line numbers and page numbers should be used in the manuscript for an optimal review process

Response: this has now been done thank you and apologies.

Reviewer 3: Mlcochova present a series of findings on the COVID Delta variant, including some epidemiological observations and some in-vitro observations. My comments are focused on the epidemiological observations.

Major comments

1. No methodology is provided for the modeling analysis which forms the motivation for the remainder of the paper. Even without a modeling analysis it is clear from the rapid growth in Delta variant outside of India that it is more transmissible than previous variants. Authors use a model here to infer that Delta is also overcoming natural immunity from prior non-Delta infections to a greater extent than other strains. Without details of the model it is not possible to appraise it. Authors included “waning of immune protection following infection, parameterised using the results of recent longitudinal cohort studies” - was this waning in antibodies or waning in protection against confirmed (re-)infection? How are you accounting in the model for heterogeneity in exposure, i.e. people such as HCWs and workers in service sector more likely to be exposed to infection in 2020 are also more likely to be exposed to infection in 2021?

Response: We apologize for this omission in our initial submission. We have now included the technical appendix describing the mathematical details of our model, which is attached in this submission. The Bayesian semi-mechanistic model that we describe is motivated by the substantial body of literature that exists developing the conceptual frameworks for understanding the dynamics and properties of pathogens that consist of multiple antigenic variants (e.g. Kucharski et al, Journal of Mathematical Biology, 2016); and has recently been used to quantify and characterise the dynamics and properties of the Gamma variant during its establishment in Manaus, Brazil (Faria et al., Science 2021). It attempts to ascribe the most plausible explanation for the data by simultaneously including and considering competing epidemiological hypotheses (such as natural waning of immunity vs immune evasion). We consider dynamics for changes in the transmissibility, and immune evasion properties of delta as compared to other circulating variants in Mumbai. The posterior distribution of parameters inferred within the modelling framework based on fitting to genomic, serological and epidemiological data shows that there is a trade-off between transmissibility and immune evasion, with the most likely situation being one of both increased transmissibility and moderate immune evasion.

The waning of immune protection that we refer to is waning of protection against re- infection. We obtain estimates from the SIREN Cohort Study in the United Kingdom ([https://www.thelancet.com/journals/lancet/article/PIIS0140-6736\(21\)00675-9/fulltext](https://www.thelancet.com/journals/lancet/article/PIIS0140-6736(21)00675-9/fulltext)) and a large-scale observational study from Denmark

([https://www.thelancet.com/journals/lancet/article/PIIS0140-6736\(21\)00575-4/fulltext](https://www.thelancet.com/journals/lancet/article/PIIS0140-6736(21)00575-4/fulltext)). Both of these studies found robust protection to infection (a relative risk of being infected having previously been infected of ~0.2 over 7+ months relative to individuals not previously infected), and limited evidence of waning over this period. Based on these results, and in conjunction with recent work highlighting that infection with SARS-CoV-2 induces antigen-specific, long-lived humoral immunity (<https://www.nature.com/articles/s41586-021-03647-4>), we therefore include limited waning of immune protection following infection, such that a fraction of 0.2 of the population of individuals become re-susceptible and available for reinfection by 7 months following infection, in line with these published results.

We thank the reviewer for raising the important point of heterogeneity in exposure. The analysis of transmissibility and cross-immunity is based on a model that uses aggregate data sources for mortality, serology and genomics for the city as a whole, rather than disaggregated by particular groups such as HCW. This is an important limitation to acknowledge, which we have clarified in the model description with the following section:

“Our model makes the assumption of a homogeneously mixed population and therefore ignores heterogeneities in contact patterns and degrees of exposures between different sub-groups such as healthcare or service workers where social distancing or working from home is not possible. Such heterogeneities would however likely concentrate new infections in those with a higher chance of having previously been infected, and therefore from this perspective, our estimates of the degree of immune-evasion are likely conservative.”

2. In the section “Transmission clusters in vaccinated health care workers”, authors hypothesised that antigenic changes were responsible for infections in healthcare workers, but did not estimate vaccine effectiveness or demonstrate that vaccine effectiveness had changed for the Delta variant.

Response: we thank the reviewer for this comment. The original aim was not to show that VE had changed for Delta, rather to demonstrate transmission amongst vaccinated HCWs.

However reviewers have raised concerns about the current cluster analysis and this will form a separate analysis. We have now performed analysis of vaccine efficacy against B.1.617.2 relative to non-B.1.617.2 as suggested by the reviewer – we think this relates more directly

to the rest of the manuscript. We are not able to assess change in VE due to lack of longitudinal data.

Ideally VE is estimated by an RCT or test negative case control study. As the former is not ethical and the latter not possible due to lack of test negative data we used an approach recently reported by PHE¹. We determined the proportion of cases with the B.1.617.2 variant relative to all other circulating variants by vaccination status. We then used logistic regression to estimate the odds ratio of testing positive with B.1.617.2 in vaccinated compared to unvaccinated individuals, adjusting for age, sex and hospital. There was an increased odds of infection with B.1.617.2 versus non- B.1.617.2 following two doses of vaccine after adjusting for age, sex and hospital. These data are presented as extended data table 4B(table below).

	B1.617.2	Non-B.1.617.2	B1.617.2: Non-B.1.617.2	OR (95% CI)	P value	aOR (95% CI)	P value
Unvaccinated	12	7	1.71	-		-	
Vaccinated							
Dose 1	23	6	3.83	2.24 (0.61-8.16)	0.22	2.06 (0.51-8.40)	0.31
Dose 2	76	7	10.86	6.33 (1.89-21.27)	0.003	5.14 (1.32-20.0)	0.018
Dose 1 and 2	99	13	7.62	4.44 (1.48-13.30)	0.008	3.59 (1.06-12.16)	0.04

Extended Data Table 4B: Relative ChAdOx-1 vaccine efficacy against B1.617.2 v non-B1.617.2: Odds ratios for detection of B.1.617.2 relative to non-B.1.617.2 in vaccinated compared to unvaccinated individuals

In Extended Data Figure 1 panels c-e the cluster sizes are larger with Delta lineage but it is not clear whether sampling was systematic, and whether a small number of larger clusters might have occurred by chance.

Response: we have now removed this analysis of clustering as explained above.

Authors did find significant differences in Ct values and please confirm that this comparison was adjusted for time since symptom onset.

Response: Time since symptom onset was similar in both groups. These data are now provided in Extended Data and we adjusted for time since onset in the analysis of Ct values. The statistical significance was lost when Ct values from hospital 2 were added following first round of reviews. This may be related to different assays being used across sites. Larger population based studies are better placed to answer the question of viral load differences for Delta.

3. Regarding the analysis of cluster sizes, in epidemiology usually a cluster is defined as 2 or more linked cases, so I'm not sure how a mean cluster size can be less than 2.

Response:

We apologise for the confusion. We were referring to sequenced infections where there was a phylogenetically linked infection in the same hospital. We have now removed cluster analysis as explained above.

Minor comments

Introduction "The B.1.617 variant emerged in the state of Maharashtra in late 2020/early2021" - are you confident it emerged here, or just that it was first detected here?

Response: we agree with this point and have amended the text.

Reviewer 4

- Summary

In this study, Mlcochova and colleagues report of the emergence of the Delta variant of SARS-CoV-2 in India and present data on Delta kinetics, immune evasion, vaccine breakthroughs, and syncytium formation. The study contains some valuable study, but the majority of the conclusions are currently not supported by the analyses performed by the authors. There is a great need to much more carefully perform the majority of the analyses in the manuscript and generally speaking focus much more on making conclusions robust and representative. For these reasons I cannot recommend publication at this stage and a major revision will be required.

Response: we thank the reviewer for this summary and we have duly made a major revision of the manuscript. We have substantially updated our Bayesian modelling analysis with background on the model, further sensitivity analyses and simulations. We have decided to remove the HCW cluster analysis given the length of the paper with the additional work we have added and the time needed to re-analyse the clustering. We have added additional replication data in the HAE system and lung cells to show conclusively that B.1.617.2 has a replication advantage over B.1.1.7 and given recent global events this is vitally important. We have also now shown with monoclonal antibodies that B.1.617.2 compromises the two clinically approved mAbs. In addition we show significant escape from NTD and RBD antibodies.

- Major points:

Their "Bayesian model of SARS-CoV-2 transmission" needs to be much better justified and additional analyses are required to establish robustness. Did the authors perform any sensitivity analyses? How dependent are they on choice of priors? How exactly did they run the model? No details have been provided in the methods.

We thank the reviewer for their comments and apologise for the omission of a technical appendix in our initial submission, which we now include with this response and which provides a full overview of the formulation of the mathematical model. We completely appreciate and agree with the reviewer of the importance of sensitivity analyses – in our initial submission, we presented sensitivity analyses surrounding 2 key inputs to the model: the degree of mortality under ascertainment and the date of B.1.617.2 emergence in Mumbai

(Extended Data Table 1A). Whilst the exact results quantitatively differ from one another, these results all support our qualitative conclusions, which are that some combination of immune evasion and increased transmissibility best explain the observed SARS-CoV-2 dynamics in Mumbai (though the comparative extent of each quantitatively differ). We completely agree with the reviewer that sensitivity of the model inference to the choices of prior is important and so have run a further set of analyses where we vary the prior on the degree of immune evasion and the extent of increased transmissibility (Extended Data Table 1B) – whilst different choices of prior change the exact degree of increased transmissibility and immune evasion inferred (as would be expected within a Bayesian framework), they do not alter the qualitative conclusions from the modelling that are presented in the main text. Finally, we now also include an additional simulation study (Extended Data Table 1C) to address the capacity of the model to jointly infer immune-evasion and increased transmissibility. We generate two synthetic datasets (available on the github link) from a hypothetical setting in which (1) a variant emerges that possesses no immune-evasion potential and is only more transmissible and (2) a variant emerges with 50% immune evasion and increased transmissibility. We then fit our modelling framework to these synthetic datasets and explore the posterior over immune-evasion and increased transmissibility inferred by the model. Our model correctly recovers the ground truth in both cases. In the case of no immune evasion, this demonstrates that our framework correctly excludes the hypothesis of immune evasion when it is not borne out by the data.

The experiments in Fig. 2a,b appear to be very preliminary. The detected amounts are extremely low (e.g. 10^{-2} / cell or 10 copies / ml) and therefore very sensitive to experimental variability. These experiments should be repeated and proper statistics applied. How are these experiments normalized to control for slight differences in measured MOIs? For the remaining experiments in this figure, why not compare to B.1.1.7? How does Delta differ compared to Alpha?

Response: The experiments have been repeated and the data are reproducible for the organoids despite low copy numbers. We have now added statistical analysis as requested. Input virus normalisation is done by input genome copy number. Moreover we have now performed replication experiments in Calu-3 lung cells and in a HAE system with an air-liquid interface, using multiple readouts of infection, with consistent results showing superior replication of B.1.617.2.

The subsequent PV data were designed to compare B.1.617.2 and B.1.617.1 to gain insight into why the former has been more successful in India despite a shared recent common ancestor. The Wuhan-1 spike is included as the control as this is the parental plasmid from which the mutants were generated. Indeed we find in some cell lines that B.1.617.2 spike confers an entry advantage compared to B.1.617.1 and that in a live virus comparison B.1.617.2 also has faster replication kinetics than B.1.617.1. Figure 2 shows that the two spikes confer similar sensitivities to neutralising antibodies following vaccination, and therefore increased infectivity for B.1.617.2 is a feasible reason for its dominance over B.1.617.1. We showed previously that Alpha had similar entry to WT in PV experiments and did not therefore seek to repeat these experiments here (Meng et al, Cell Reports 2021).

"We found that B.1.617.2 was marginally less sensitive to the TMPRSS2 inhibitor Camostat"
 - proper stats need to be applied here and a statistical statement made.

Response: The difference was modest but significant though we have removed the camostat data as live virus experiments did not show the same result and significant further investigation beyond the scope of this paper is needed to understand the mechanisms.

	WT	617.1	617.2
IC50	5.046	5.936	12.95
IC50 shift (95% CI)	-	1.152 (0.8637 to 1.554)	3.154 (2.088 - 4.825)

The whole "Transmission clusters in vaccinated health care workers associated with Delta variant" needs to be reanalyzed. Sequences are not independent and it's impossible to distinguish transmission within a cluster vs those sourced from the community by using criteria such as "we defined related or 'linked' infections as differing by six nucleotides or less". This is especially true since B.1.617.2 is overall more diverse than e.g., B.1.1.7 and has a different history, hence conclusions about "mean cluster size" end up being fraught with

problems. This whole section will require significant additional analyses and a rewrite - it's very unclear what exactly the authors did and what can be concluded based on the data and analyses. I don't doubt that the authors may be correct in some of their conclusions, but the current set of data and analyses are insufficient to support the conclusions made in this section. In the abstract it's also mentioned that "Delta variant [...] dominates vaccine-breakthrough infections", but it's unclear if this is simply due to its prevalence or if it's because it's somehow better able than e.g., alpha in causing vaccine breakthroughs. That is not clear from the present study.

Response: We have decided to remove the HCW clustering analysis. We have added a relative vaccine effectiveness analysis in HCW comparing B.617.2 to non-B.617.2 as suggested by another reviewer. This analysis is more directly related to the themes of immune evasion and transmissibility in the rest of the paper.

"The B.617.1 and B.617.2 spikes demonstrated higher fusion activity and syncytium formation, mediated specifically by P681R" The authors have previously done these experiments for alpha - how do they compare?

Response: we now show in the revised manuscript that fusion kinetics for B.1.1.7 spike are similar to B.617.1 and B.617.2 spikes.

All data should be made available immediately on GISAID. The authors use publicly available data (via GISAID), but have not themselves made their data available - that is not acceptable.

Response: the HCW sequences were available via the github link at submission. There was an issue with GISAID upload in India and this should be completed very soon.

The entire discussion section needs work. It's currently very confusing, meanders through different findings, and contains several half-sentences, missing references, and run-on sentences.

Response: we have now significantly modified the discussion and added references throughout.

- Minor points:

Typos and inconsistencies throughout. E.g., from abstract "ChadOx-1" should either be "ChAdOx1" or "AZD1222"; "HCW", "GMT" are undefined; "shows 8 fold approximately reduced sensitivity" should be "shows approximately 8-fold reduced sensitivity", etc.

Response: we thank the reviewer for pointing these out and have amended the text

Introduction about the epidemic in India should be updated to reflect current situation and careful attention should be given to properly introducing the background and questions - it's a bit of a mess at the moment. Referencing all needs to be improved and the usage of "California" variant should be avoided (the WHO has designated this variant as "Epsilon").

Response: we have now amended the text to reflect the changing situation and stated the question to be addressed in the work.

- 1 Bernal, J. L. *et al.* Effectiveness of COVID-19 vaccines against the B.1.617.2 variant. *medRxiv*, 2021.2005.2022.21257658, doi:10.1101/2021.05.22.21257658 (2021).

Reviewer Reports on the First Revision:

Referees' comments:

Referee #1 (Remarks to the Author):

The authors, while addressing several comments, have now removed the most interesting part of the paper - the analysis of B.1.617.2 clusters in HCWs.

In addition, the assumption of a 0.05-0.1 IFR is not plausible given the IFRs measured elsewhere (which reaches 1% or higher in some localities).

Referee #2 (Remarks to the Author):

Figure 3 would be improved by adding in some written descriptors along with the a, b, c) etc. – for instance it is not immediately clear what the difference is between panels a) and b) – also the order of panels a-f) is a bit confusing. Examples would be to add the “Calu3” heading to panel v, and “pseudoparticles” and “infectious virus” to panels h, I and j)

For Figure 4, it is clear that “WT” is lower, are there any statistically differences between the variants in panels d) and e)

For Figure 4, B.1.1.7 should be added to panel c)

For Figure 4, the data on CMK is off-topic and may lead to suggestions on use furin-inhibitors as therapeutics, with these much-removed from useful clinical trials – studies on camostat were removed, and this reviewer recommends CMK data to also be removed

For Figure 4, adding in a DAPI stain would help visualize syncytia, which should also be counted in some way (even if not highly quantitative)

Referee #3 (Remarks to the Author):

I went through the responses to reviewers and the revisions that have been made

Authors now focus on the following:

(1) a reduction in neutralising antibodies to Delta -- important but not novel, many other published studies have reported this. The loss of sensitivity to monoclonal antibodies is also noted but perhaps less important given the limited clinical use of these.

(2) increased replication of Delta in vitro -- noted

(3) modeling analysis -- I still can't find the mathematical details in the supplementary information. Authors clarify that their estimates are likely conservative, which I could suggest to rephrase as "likely biased". The modeling study could be removed from this manuscript.

(4) reduced VE in HCWs in India -- would consider anecdotal evidence, not a systematic study, except for the final comparison where the adjusted odds ratio for B.1.617.2 relative to non-B.1.617.2 was 5.14 (95% CI 1.32-20.0, p=0.018). This is interesting and an important observation, although from a very simplistic study design and simplistic analysis.

Referee #4 (Remarks to the Author):

In the revised manuscript, Mlcochova and colleagues have made various revisions, removed significant data, and added some new experimental data. While the revisions help improve aspects of parts of the manuscript, I remain unconvinced by some of the major conclusions and I feel that there's a need for a more thorough and careful study, as opposed to a rushed one.

Specifically, some of the sections are not novel, however, more importantly in other sections I have concerns about the robustness of the conclusions due to rushed analyses, biases in data, small effect sizes, or too small data sets.

The following sections are not novel, but may add some useful information - although I think it's time that we reassess whether continuing to publish purely in vitro studies of Ab binding/neutralization can still be justified:

B.1.617.2 shows reduced sensitivity to neutralising antibodies from recovered individuals

B.1.617.2 shows reduced sensitivity to vaccine-elicited antibodies

B.1.617.2 is less sensitive to monoclonal antibodies (mAbs)

The following sections do not appear to have robust conclusions or have small effect sizes:

B.1.617.2 growth advantage due to re-infection and increased transmissibility

B.1.617.2 variant shows higher replication in human airway model systems

B.1.617.2 spike has enhanced entry efficiency associated with cleaved spike

Breakthrough B.1.617.2 infections in ChAdOx-1 vaccinated health care workers

While the authors now explain their Bayesian model in some detail, given the biases and uncertainties, I remain unconvinced by its utility. The model itself - while it has been published previously - needs to be much more carefully scrutinized and validated. While I appreciate that it is possible to separate time-varying effects from constant effects of immunity (the former) and transmission (the latter), there is so much uncertainty in understanding e.g., immune waning that I am not convinced that the results produced are meaningful.

The "breakthrough" investigations are preliminary, and while I don't doubt that the authors may well be correct, I feel much more in-depth studies are required. The authors removed the "HCW clusters" part of the paper, which I previously described as preliminary as well. I feel that in this particular case, instead of rushing through publication, all these epi studies should be more carefully conducted, combined, and analyses expanded - there would be much more value in such an approach (e.g., see recent B.1.1.7 studies from the COG-UK team).

The section on "B.1.617.2 spike confers increased syncytium formation" is of interest, but it's unclear how important these findings are to human pathophysiology of COVID-19.

Overall, while there is some value-add from this study, I feel that it is rushed and the main conclusions remain preliminary. However, I appreciate that the authors might well end up being correct - not because the data and analyses as they are lead me to that conclusion, but more because of what we have learned about SARS-CoV-2 variants over the last six months . Given the rush to publishing on the latest and greatest variant can be important, my personal opinion is that there needs to be a renewed focus on more complete studies that are well-supported based on extensive datasets. This particular study is not an example of that, although I fully understand if Nature might think differently.

Referee #5 (Remarks to the Author):

Within this paper, the authors analyze outbreaks at three health facilities in Delhi. Though the majority of symptomatic cases captured were B.1.617.2, some were from other lineages. The authors did not have access to test negative data to construct a test-negative analysis. Potentially they have access to staff employment and vaccination records, and could construct a cohort analysis to estimate absolute effectiveness. But the analysis presented uses an approach described by Lopez-Bernal in medRxiv to estimate relative effectiveness of the vaccine against B.1.617.2 versus other lineages.

The main approach of the analysis is reasonable, but there are several important limitations. First, the authors have not specified details of the analysis. It does not even seem to be listed in the statistical methods section? How is age modeled, for example?

Second, the model does not address calendar time as a potential confounder. Namely, we know there is an increase in the prevalence of B.1.617.2 over time, and we know vaccination coverage increases over time. Maybe vaccination coverage is fairly steady over time within health facilities, in which case it will not be an important confounder, but this is not addressed.

The final limitation is that the sample sizes are small, especially for the non-B.1.617.2 lineages, and the available confounders are limited. This should be more clearly highlighted as a limitation/caveat in the interpretation. It also raises points about model sensitivity and robustness.

What methods were used to check model fit?

Line 349. Effectiveness instead of efficacy. This is in many places throughout.

Line 394. Odds of symptomatic infection or disease, instead of infection.

Author Rebuttals to First Revision:

Reviewer 1

The authors, while addressing several comments, have now removed the most interesting part of the paper - the analysis of B.1.617.2 clusters in HCWs.

Response: we thank the reviewer for appreciating the responses to previous comments and the additional work we have undertaken. Although the analysis of clusters is interesting, our HCW data in the paper show a number of important and interesting aspects that are no less important than the clustering:

- 1. A super spreader or 'overdispersion' event occurred in hospital 1, with 10 vaccinated HCW being infected with almost identical viruses (the first such recorded event in vaccinated individuals).*
- 2. We provide clinical and demographic characteristics of over 130 breakthrough infections in HCW, including Ct values that indicate high viral loads in infected HCW, genomic information, detailed vaccination status and clinical severity. This is a unique dataset and resource for the scientific community and will permit further valuable analysis.*
- 3. We also now show additionally with a vaccine effectiveness (VE) analysis that the ChAdOx-1 vaccine is less effective against Delta virus in vaccinated HCW compared to non-Delta virus. These are unique data given that few other settings have had high transmission pressure from a mix of variants in order to be able to perform such a study. Reviewers 3 and 5 agree with the importance of this finding.*

In addition, the assumption of a 0.05-0.1 IFR is not plausible given the IFRs measured elsewhere (which reaches 1% or higher in some localities).

Response: there has regrettably been confusion here on the part of the reviewer, but we have now removed the analysis in question.

Reviewer 2

Figure 3 would be improved by adding in some written descriptors along with the a, b, c) etc. – for instance it is not immediately clear what the difference is between panels a) and b) – also the order of panels a-f) is a bit confusing. Examples would be to add the “Calu3” heading to panel v, and “pseudoparticles” and “infectious virus” to panels h, I and j)

Response: We thank the reviewer for this very helpful suggestion and have done this

For Figure 4, it is clear that “WT” is lower, are there any statistically differences between the variants in panels d) and e)

Response: There was no statistical difference between variants.

For Figure 4, B.1.1.7 should be added to panel c)

Response: We thank the reviewer for this very helpful suggestion and have now add B.1.1.7 to the western blot.

For Figure 4, the data on CMK is off-topic and may lead to suggestions on use furin-inhibitors as therapeutics, with these much-removed from useful clinical trials – studies on camostat were removed, and this reviewer recommends CMK data to also be removed

Response: We agree and have removed the CMK data panel

For Figure 4, adding in a DAPI stain would help visualize syncytia, which should also be counted in some way (even if not highly quantitative)

Response: DAPI reading cannot be done unfortunately as the Incucyte instrument used for analysis does not read DAPI. In terms of counting, we do indeed count the phase area, which is the number of cells that are in the well. Furthermore, the syncytia are only formed when two split-GFP cells fuses. The % plotted in the graph is the ratio between Green Area(GFP)/

total Cell area (which marks the cells) so the number of syncytia are already normalised to the amount of cells present in the well. These details are in the methods.

Reviewer 3:

I went through the responses to reviewers and the revisions that have been made

Authors now focus on the following:

(1) a reduction in neutralising antibodies to Delta -- important but not novel, many other published studies have reported this. The loss of sensitivity to monoclonal antibodies is also noted but perhaps less important given the limited clinical use of these.

Response: we thank the reviewer for this comment. Indeed there are three published studies for Delta and whole live virus neutralisation (one as a correspondence). However details are important: Ours is the only to perform both live virus and pseudotyped virus work, and importantly our associated pre-print was the first to show a lower neutralisation titre for ChAdOx-1 compared to BNT162b2, now also being reported in other pre-prints. We have also uniquely compared neutralisation sensitivity B.1.617.2 (Delta) with B.1.1.617.1 in the same manuscript using a panel of over 60 vaccine sera following two vaccine platforms, and the importance of this is to understand the dominance of the former over the latter. We find that the two variants have similar sensitivity to the vaccine elicited sera. Coupled with the data showing higher replication of Delta over B.1.617.1 we can conclude that the dominance is primarily driven by higher infectivity. This is very important as we go forward and determine the mutations responsible for these phenotypic differences.

On the point of monoclonals we agree that at global public health level therapeutic mAb at present have had little role. However, they could be made at low cost and combined with steroid therapy. The vaccination rates globally are very low and severe infection is leading to significant morbidity and mortality. mAb could address this and our data on Delta and mAb are therefore highly relevant. In addition, availability of therapeutic and possibly prophylactic mAb will be critical for the millions of patients worldwide who have poor vaccine responses due to immune suppression.

(2) increased replication of Delta in vitro -- noted

(3) modeling analysis -- I still can't find the mathematical details in the supplementary information. Authors clarify that their estimates are likely conservative, which I could suggest to rephrase as "likely biased". The modeling study could be removed from this manuscript.

RESPONSE: we have now removed this section as suggested.

(4) reduced VE in HCWs in India -- would consider anecdotal evidence, not a systematic study, except for the final comparison where the adjusted odds ratio for B.1.617.2 relative to non-B.1.617.2 was 5.14 (95% CI 1.32-20.0, $p=0.018$). This is interesting and an important observation, although from a very simplistic study design and simplistic analysis.

RESPONSE: Yes we agree that this is a very significant piece of information from early part of the Delta epidemic in India. The findings are highly relevant for limiting HCW associated COVID-19 globally. Highly complex analyses suffer their own limitations and ours is from a unique and well characterised cohort of HCW with detailed genomic, lab and clinical outcome data. Also one should note that our approach has been used by public health bodies for a paper recently published in NEJM (Bernal et al).

Reviewer 4

In the revised manuscript, Mlcochova and colleagues have made various revisions, removed significant data, and added some new experimental data. While the revisions help improve aspects of parts of the manuscript, I remain unconvinced by some of the major conclusions and I feel that there's a need for a more thorough and careful study, as opposed to a rushed one.

Response: the only significant data removed were on the clustering and the modelling. The modelling and clustering is being more thoroughly analysed for separate pieces of work.

Specifically, some of the sections are not novel, however, more importantly in other sections I have concerns about the robustness of the conclusions due to rushed analyses, biases in

data, small effect sizes, or too small data sets.

The following sections are not novel, but may add some useful information - although I think it's time that we reassess whether continuing to publish purely in vitro studies of Ab binding/neutralization can still be justified:

B.1.617.2 shows reduced sensitivity to neutralising antibodies from recovered individuals

B.1.617.2 shows reduced sensitivity to vaccine-elicited antibodies

B.1.617.2 is less sensitive to monoclonal antibodies (mAbs)

Response: our group was among the first to pre-print the above data and as outlined above we did both PV and live virus in addition to comparing with B.1.617.1, something not done to date by others in the same manuscript. We agree studies on pure neutralisation are very limited and that is why we have presented this cross disciplinary, collaborative and extensive piece of work.

The following sections do not appear to have robust conclusions or have small effect sizes:

B.1.617.2 growth advantage due to re-infection and increased transmissibility

B.1.617.2 variant shows higher replication in human airway model systems

B.1.617.2 spike has enhanced entry efficiency associated with cleaved spike

Breakthrough B.1.617.2 infections in ChAdOx-1 vaccinated health care workers

Response:

1. The Bayesian analysis has now been removed.

2. The higher replication with live virus shows differences that are 10-100 fold and for PV 10 fold. This is robust as in three tissue model systems with two virus model systems.

3. The breakthrough and VE data are clear and n=130 is not small given the granularity of data including vaccination dates, symptoms, Ct values, clinical severity.

While the authors now explain their Bayesian model in some detail, given the biases and uncertainties, I remain unconvinced by its utility. The model itself - while it has been published previously - needs to be much more carefully scrutinized and validated. While I appreciate that it is possible to separate time-varying effects from constant effects of

immunity (the former) and transmission (the latter), there is so much uncertainty in understanding e.g., immune waning that I am not convinced that the results produced are meaningful.

Response: this section has been removed.

The "breakthrough" investigations are preliminary, and while I don't doubt that the authors may well be correct, I feel much more in-depth studies are required. The authors removed the "HCW clusters" part of the paper, which I previously described as preliminary as well. I feel that in this particular case, instead of rushing through publication, all these epi studies should be more carefully conducted, combined, and analyses expanded - there would be much more value in such an approach (e.g., see recent B.1.1.7 studies from the COG-UK team).

Response: we appreciate this comment. However, more detailed epi analysis would require more detail on HCW movements and patient diagnoses. We do not have access to such data, unlike COG-UK where there are detailed in-hospital studies. Our data on reduced VE against Delta v non Delta are critically important and have already informed the latest CDC guidance on mask wearing. Those data were done using a well-known and robust method and the results are in keeping with a recent PHE NEJM paper in the community. The NEJM study however lacks the fine detail of our dataset.

The section on "B.1.617.2 spike confers increased syncytium formation" is of interest, but it's unclear how important these findings are to human pathophysiology of COVID-19.

Response: it is true we do not fully understand the physiology but an important and novel result nonetheless, with all the relevant controls and the demonstration that Delta mediated fusion is less sensitive to neutralising antibodies.

Overall, while there is some value-add from this study, I feel that it is rushed and the main conclusions remain preliminary. However, I appreciate that the authors might well end up being correct - not because the data and analyses as they are lead me to that conclusion, but more because of what we have learned about SARS-CoV-2 variants over the last six months . Given the rush to publishing on the latest and greatest variant can be important, my personal opinion is that there needs to be a renewed focus on more complete studies that are well-

supported based on extensive datasets. This particular study is not an example of that, although I fully understand if Nature might think differently.

Response: we appreciate this view, but we have aimed to provide a multi-dimensional piece of work that explains our current situation and can therefore inform policy at the highest level during a pandemic that is far from over.

Reviewer 5

Within this paper, the authors analyze outbreaks at three health facilities in Delhi. Though the majority of symptomatic cases captured were B.1.617.2, some were from other lineages. The authors did not have access to test negative data to construct a test-negative analysis. Potentially they have access to staff employment and vaccination records, and could construct a cohort analysis to estimate absolute effectiveness. But the analysis presented uses an approach described by Lopez-Bernal in medRxiv to estimate relative effectiveness of the vaccine against B.1.617.2 versus other lineages.

Response: we indeed used a similar approach to Bernal et al (now published in NEJM), validated against a test negative method in that paper with very good concordance. We are unlikely to be able to set up the cohort analysis suggested in a timely way due to inefficiencies within the Indian hospital system.

The main approach of the analysis is reasonable, but there are several important limitations. First, the authors have not specified details of the analysis. It does not even seem to be listed in the statistical methods section? How is age modeled, for example?

Response: we have now provided extensive further details of the analysis in the methods. Age was modelled as a continuous variable.

Second, the model does not address calendar time as a potential confounder. Namely, we know there is an increase in the prevalence of B.1.617.2 over time, and we know vaccination coverage increases over time. Maybe vaccination coverage is fairly steady over time within health facilities, in which case it will not be an important confounder, but this is not addressed.

Response: the infections were documented within a month window and therefore the time dependent effects of vaccination are not important in this study. We have now noted this in the results text as suggested and thank the reviewer for this comment.

The final limitation is that the sample sizes are small, especially for the non-B.1.617.2 lineages, and the available confounders are limited. This should be more clearly highlighted as a limitation/caveat in the interpretation.

Response: we have now addressed these limitations in the results text as suggested: 'Calendar time, often associated with vaccination status, was unlikely to be a significant confounder here given the short time period studied. The analysis presented, whilst limited by relatively small numbers of non-B.1.617.2 infections and potentially affected by unmeasured confounders, is nevertheless consistent with UK data where the non-B.1.617.2 infections were largely B.1.1.7.'

It also raises points about model sensitivity and robustness. What methods were used to check model fit?

*Response: The final regression model was adjusted for age as a continuous variable, sex and hospital. Model sensitivity and robustness to inclusion of these covariates was tested by an iterative process of sequentially adding the covariates to the model and examining the impact on the ORs and confidence intervals until the final model was constructed (**Extended Data Table 4**). The R-square measure, as proposed by McFadden⁵⁰, was used to test the fit of different specifications of the same model regression. This was done by sequential addition of the variables adjusted for including age, sex and hospital until the final model was constructed. In addition, the absolute difference in Bayesian Information Criterion (BIC) was estimated. The McFadden R² measure of final model fitness was 0.11 indicating acceptable model fit. The addition of age, gender and hospital in the final regression model improved the measured fitness. However, the absolute difference in BIC was 13.34 between the full model and the model excluding the adjusting variable, providing strong support for the parsimonious model. The fully adjusted model was nonetheless used as the final model as the sensitivity analyses (**Extended Data Table 4**) showed robustness to the addition of the*

covariates. The OR for B.1.617.2vs non-B.1.617.2 attenuates slightly with the addition of more covariates, but it remains significant and all confidence intervals overlap.

Line 349. Effectiveness instead of efficacy. This is in many places throughout.

Response: we have replaced efficacy with effectiveness

Line 394. Odds of symptomatic infection or disease, instead of infection.

Response: we have corrected this and thank the reviewer

Reviewer Reports on the Second Revision:

Referees' comments:

Referee #5 (Remarks to the Author):

Thank you to the authors for their clarifications and edits. I have a few additional questions.

1. Regarding temporal confounding, the study took place from late March to early May. The authors reply that time is not considered because the period is very short, but as we have seen repeatedly with this variant, delta prevalence can very quickly change. From examination of Figure 5, non-delta variants tend to occur earlier. Also, given the presence of some individuals with one vaccine dose, the vaccination campaign is ongoing over time. While time will not necessarily have a large impact on the results, I would prefer that the authors provide direct evidence of robustness.

2. When I noted model-checking in my original review, I was referring to the functional form of the covariates as well as what covariates were included. For example, was there any checking of the suitability of age as a continuous variable? Versus a spline, for example. The key is really – are the conclusions regarding relative VE robust to assumptions about the functional form and set of covariates included.

3. Line 309-312. This sentence is not clear as written. "Consistent with UK data where... reduced effectiveness against the delta variant was observed"?

4. Abstract has not been updated to use the appropriate terminology of vaccine effectiveness, and to temper the conclusions considering the limitations.

5. Line 339. "symptomatic infection and disease" is redundant.

Author Rebuttals to Second Revision:

1. The title is over the limit at 111 characters.

Response: this has now been corrected

2. Subheadings need to be reduced to 40 characters.

Response: this has now been done

3. Figures 1, 3, 4 and 5 are too tall (over 18cm).

Response: this has now been corrected

4. The paper exceeds the length limit of 6 pages, as well as the limit of display items-please edit the manuscript and consolidate display items to 4 Main items, moving other information to the Extended Data. Please provide a detailed Author Contribution statement that comprises all authors and all approaches in the manuscript.

Response: this has now been done.

• Please make sure that competing interests are properly declared in the manuscript and EPC, for all authors.

Response: this has now been corrected

• Please ensure that human data is reported properly, including statements of informed consent acquisition for all samples.

Response: this has now been done

• As discussed with Dr Thomas, please deposit all the data on approved repositories, and provide all accession numbers. The data must be available before we can proceed with publication.

Response: this has now been done and accession numbers are live

Referee #5 (Remarks to the Author):

Thank you to the authors for their clarifications and edits. I have a few additional questions.

1. Regarding temporal confounding, the study took place from late March to early May. The authors reply that time is not considered because the period is very short, but as we have seen repeatedly with this variant, delta prevalence can very quickly change. From examination of Figure 5, non-delta variants tend to occur earlier. Also, given the presence of some individuals with one vaccine dose, the vaccination campaign is ongoing over time. While time will not necessarily have a large impact on the results, I would prefer that the authors provide direct evidence of robustness.

Response: we have now provided direct evidence of robustness as follows: We tested whether the odds ratio for B.1.617.2 vs non-B.1.617.2 lineages significantly alters under different model parameterizations. We tested three models of increasing degrees of complexity. First, we tested a linear model with a linear effect for week to account for unobserved linear changes in time. The resultant odds ratio from this model was 6.68 [1.71-27.39]. We then generalised the weekly effect to be discrete independent identically distributed effects for each week, for which the resultant odds ratio was 6.49 [1.6,28.8]. Therefore the results are highly robust to the addition of time and we have added text to reflect this, as well as addition of these estimates to the extended data table 4 (last 2 rows of bottom table).

2. When I noted model-checking in my original review, I was referring to the functional form of the covariates as well as what covariates were included. For example, was there any checking of the suitability of age as a continuous variable? Versus a spline, for example. The key is really – are the conclusions regarding relative VE robust to assumptions about the functional form and set of covariates included.

Response: we have now modelled age as a continuous versus spline with no impact on results. We tested a linear model with a B-Spline basis for age (with 4 degrees of freedom) where the resultant odds ratio was 4.49 (1.38-14.48). These more complex models all consistently suggest that a significant odds ratio for B.1.617.2 vs non B.1.617.2 is not a modelling artefact or heavily dependent on model structure. We have indicated this in the revised text and added a further row for age as spline in ext data table 4.

3. Line 309-312. This sentence is not clear as written. “Consistent with UK data where... reduced effectiveness against the delta variant was observed”?

Response: we have removed this sentence as it was unclear.

4. Abstract has not been updated to use the appropriate terminology of vaccine effectiveness, and to temper the conclusions considering the limitations.

Response: we have now amended terminology and tempered the conclusions with limitations as requested.

5. Line 339. "symptomatic infection and disease" is redundant.

Response: we have now removed this as requested.